# DiP-G: Discrete Prompting for Graph Neural Networks

Yumeng Zhao [1]   Huiying Hu [2]   Shuo Wen [3]   Junjie Shen [1]   Bei Hua [1]

## Abstract

Graph Neural Networks (GNNs) are increasingly adopting the "pre-training, adaptation" paradigm, which first pre-train GNNs on large-scale unlabeled graph data and then adapt them to specific downstream tasks. As a common pattern, graph prompting adapts to the frozen encoder by modifying the input graph structure, rather than fine-tuning the model parameters. However, most existing graph prompting approaches optimize the continuous and weighted adjacency structure in the adaptation phase, while requiring a hard discretization at inference time. This difference causes a train-test mismatch which is particularly harmful in few-shot regimes. To address the issue, we propose **Di**screte **P**rompting for **G**raphs, a discrete prompting framework that directly learns task-specific topology prompts in the combinatorial space. DiP-G operates on multi-hop local candidate subgraphs to ensure scalability, generates hard (k)-sparse prompts through a perturbed Top-(k) solver, and optimizes the discrete structures using an I-MLE gradient estimator. To improve the efficiency of backward pass, we further introduce an adaptive active-set screening rule that accelerates the target solve and can provably maintain the accuracy of the solution. Extensive experiments conducted on multiple benchmark datasets have validated the effectiveness of our proposed method.

## 1. Introduction

Graphs provide a natural representation of relational data in a wide range of fields, including social and information networks (Wang et al., 2025; Wei et al., 2023; Zhou et al., 2023b;a), knowledge graphs (Wang et al., 2024), recommender systems(Zhao et al., 2025), biology (Kang et al., 2022) and others. As a powerful tool for modeling graph data, Graph Neural Networks (GNNs)(Chen et al., 2020; Hamilton et al., 2017; Kipf, 2016) show strong performance in tasks like node classification (Luan et al., 2023; Zhao et al., 2021) and link prediction (Zhang & Chen, 2018; Zhu et al., 2024). However, how to deploy GNN in realistic scenarios remains a challenge. Supervised training requires substantial labeled data which is usually scarce, and the generalization performance of models trained for a specific task or dataset is poor when migrating to a new task.

To alleviate these issues, a growing line of work adopts the *pre-training, adaptation* paradigm, where a graph encoder is first pre-trained on large-scale unlabeled graphs based on self-supervised objectives, and is then adapted to label-scarce downstream tasks (Fang et al., 2023; Huang et al., 2024; Liu et al., 2023; Sun et al., 2022; 2023; Fu et al., 2025a;b; Yu et al., 2024; Zhili et al., 2024). The core difficulty in this pipeline lies in the adaptation stage, which must make full use of pre-trained knowledge while avoiding overfitting to a small support set. Inspired by recent prompt tuning approaches in natural language processing (Khattak et al., 2023; Zhou et al., 2022) and computer vision (Jia et al., 2022; Yoo et al., 2023), graph prompting (Fu et al., 2025c) offers an appealing approach for this purpose. Instead of fine-tuning the entire encoder, graph prompting keeps the pre-trained GNN models frozen and learns a task-specific modification of the input graph, which is a lightweight and generally more stable adaptation mechanism (Yu et al., 2025; Fatemi et al., 2021; Liu et al., 2022; Zhang et al., 2023).

Although considerable progress has been made, existing graph prompts are still a key practical limitation. Most models perform continuous relaxation optimization on the adjacency matrix in the adaptation phase, while discrete graphs are required for testing, which will lead to train–test mismatch, as shown in Figure 1. This difference is important because the pre-trained encoder usually learns from the discrete graph, and the mismatch changes the behavior of message passing, leading to the instability of optimization, especially in few-shot scenarios. Therefore, it is natural to ask such a question: how can we design a framework to directly and accurately learn graph prompts on discrete structures?

[1]School of Computer Science and Technology, University of Science and Technology of China, Hefei, China [2]Wangxuan Institute of Computer Technology, Peking University, Beijing, China [3]School of Computer Science, McGill University, Montreal, Canada. Correspondence to: Bei Hua <bhua@ustc.edu.cn>.

*Proceedings of the 43^{rd} International Conference on Machine Learning*, Seoul, South Korea. PMLR 306, 2026. Copyright 2026 by the author(s).

## Discretization Gap

Relaxed(train)      Discrete(infer)

$$\mathbf{A}_\phi \in [0,1] \qquad \widehat{\mathbf{A}} = \mathcal{D}(\mathbf{A}_\phi) \in \{0,1\}$$

$$\triangle \quad \mathbb{E}_{\text{train}}\big[ f(\mathbf{A}_\phi, \mathbf{X}) \big] \neq f(D(\mathbf{A}_\phi), \mathbf{X})$$

*Figure 1.* Train–test mismatch in relaxed graph prompting.

To address the above issue, we propose **DiP-G** (**Di**screte **P**rompting for **G**raphs), a graph prompting framework that learns task-specific prompted structures directly in the combinatorial space. In DiP-G, we operate on multi-hop local candidate subgraphs and score candidate edges on cached backbone embeddings via a lightweight parametric prompt module to ensure scalability. We also generate hard $k$-sparse discrete prompts with a perturbed Top-$k$ solver. Due to the discreteness and non-differentiability of the solver, DiP-G adopts an I-MLE estimator to obtain practical finite-difference gradient signals for prompt learning. In the end, in order to improve the efficiency of back propagation, we design an adaptive active-set screening rule that accelerates target solve and can be provably exact under a simple margin condition. By keeping the discreteness of the prompted graph throughout the training and inference, DiP-G aligns the input in the adaptation phase with the discrete domain during pre-training. The experimental results validate the superiority of DiP-G in achieving a more stable and efficient few-shot adaptation.

We summarize our main contributions as follows:

- **Discretization gap.** We identify a core mismatch in graph prompting caused by relaxed adjacencies during adaptation and discretized graphs at inference, which is particularly harmful in few-shot settings.

- **DiP-G design.** We propose DiP-G, which directly learns *hard* $k$-sparse graph prompts in the combinatorial space via a perturbed Top-$k$ solver on local candidate subgraphs, and optimizes them with an I-MLE estimator integrated with an adaptive screening routine.

- **Theoretical analysis.** We provide theoretical analysis of the conditions for the exact screened target solve, and analyze the reduction in the backward-pass cost.

- **Empirical results.** Extensive experiments conducted on multiple datasets and pre-trained backbone verify the effectiveness and stability of our DiP-G.

**Conflict of Interest Disclosure.** The authors declare that they have no financial conflicts of interest related to this work. In particular, this paper does not evaluate a model, product, service, or system developed by a company employing any of the authors.

## 2. Related Work

### 2.1. Graph Pre-training

Graph pre-training aims to learn transferable graph encoders from large-scale unlabeled graphs in a self-supervised fashion(Wu et al., 2021). A common line of work is contrastive pre-training(Sun et al., 2019), which aims to learn high-quality graph embeddings by maximizing the mutual information(MI) between two augmented graph instances. For example, DGI (Veličković et al., 2018) maximizes the MI objective between node-level representations and a global summary signal. More recent contrastive frameworks such as SimGRACE (Xia et al., 2022) and GraphCL (Hafidi et al., 2020) instantiate this principle by constructing two perturbed views through feature masking and edge perturbation, and optimizing agreement at the node or subgraph level. In parallel, generation-based pre-training has gained traction by masking parts of the input and training the model to reconstruct them. GraphMAE is a representative masked autoencoding approach tailored for graphs (Hou et al., 2022). These pre-trained encoders provide strong inductive biases, but adapting them to label-scarce tasks without full fine-tuning remains challenging in practice.

### 2.2. Graph Prompting for Adaptation

Graph prompting has emerged as an efficient way to adapt pre-trained GNN encoders. In graph prompting, the backbone stays frozen and lightweight prompts are learned for a downstream objective. Most existing approaches focus on feature-side intervention. GPPT (Sun et al., 2022) reformulates node classification as link prediction to better match common pre-training objectives. GraphPrompt and GraphPrompt+ (Liu et al., 2023) learn task prompts that reweight node features to unify pre-training and downstream prediction. GPF and its stronger variant GPF-plus (Fang et al., 2023) introduce universal prompt vectors that can be attached to input features, and MultiGPrompt (Yu et al., 2024) further injects prompts across multiple hidden layers to enhance expressivity. All-in-One (Sun et al., 2023) studies prompt-based adaptation across heterogeneous downstream tasks under a unified formulation. More recently, ProNoG (Yu et al., 2025) targets non-homophilic settings by designing prompt learning for heterophily patterns. Be-

yond feature prompts, several works show that editing the message-passing channel can be equally important. Graph-TOP (Fu et al., 2025b) works as a topology-oriented prompting framework to adapt pre-trained GNN models by modifying the graph topology of the input graph, while EdgePrompt and EdgePrompt+ (Fu et al., 2025a) adopt edge-level prompt that directly modulates how information flows across edges. Although relaxation-based edge prompting is convenient to optimize, these methods learn prompts on continuous structures but are evaluated on sparse discrete graphs, which can be particularly fragile in few-shot settings.

## 3. Preliminaries

### 3.1. Problem Formulation

We consider adapting a pre-trained graph encoder to a downstream node classification task under limited supervision. Let $\mathcal{G} = (\mathcal{V}, \mathcal{E}, \mathbf{X})$ be an attributed graph with $N = |\mathcal{V}|$ nodes and feature matrix $\mathbf{X} \in \mathbb{R}^{N \times d}$, and let $\mathbf{A} \in \{0, 1\}^{N \times N}$ denote its adjacency matrix.

We assume access to a pre-trained GNN encoder $f_{\theta^*} : (\mathbb{R}^{N \times N}, \mathbb{R}^{N \times d}) \to \mathbb{R}^{N \times h}$. Following the standard pre-training–adaptation protocol, we freeze $\theta^*$ during downstream adaptation so that the transferable representations learned from large-scale objectives (e.g., link prediction) are preserved.

A downstream task $\mathcal{T}$ is formulated as a $C$-way $K$-shot node classification problem. Let $\mathcal{S}_{\text{supp}}$ be a support set containing $K$ labeled nodes per class, and $\mathcal{Q}$ be a query set for evaluation. The goal of adaptation is to learn a task-specific prediction head $g_\omega : \mathbb{R}^h \to \mathcal{Y}$ together with a task-dependent graph prompt method that modifies the input structure, so that the induced model $g_\omega \circ f_{\theta^*}$ achieves low task loss at $\mathcal{Q}$ under supervision of $\mathcal{S}_{\text{supp}}$.

### 3.2. The Discretization Gap Analysis

A common design in graph prompting methods is to optimize graph structure through a continuous surrogate. They introduce a learnable relaxed adjacency $\mathbf{A}_\phi \in [0, 1]^{N \times N}$, typically via Gumbel-Softmax or Bernoulli-type relaxations, and optimize

$$\min_{\phi, \omega} \mathcal{L}_{task} \left( g_\omega(f_{\theta^*}(\mathbf{A}_\phi, \mathbf{X})) \right). \qquad (1)$$

However, the downstream model operates on a discrete graph at inference time. Let $\mathcal{D}(\cdot)$ denote a discretization operator that maps the relaxed matrix to a binary adjacency $\hat{\mathbf{A}} = \mathcal{D}(\mathbf{A}_\phi) \in \{0, 1\}^{N \times N}$. This gap leads to a mismatch between the structure used for optimization and inference. In particular, due to training-time randomness from the relaxation and mini-batch sampling, the encoder representa-

tions satisfy

$$\mathbb{E}_{\text{train}}[f_{\theta^*}(\mathbf{A}_\phi, \mathbf{X})] \neq f_{\theta^*}(\text{discretize}(\mathbf{A}_\phi), \mathbf{X}), \qquad (2)$$

where $\mathbb{E}_{\text{train}}[\cdot]$ denotes expectation over training-time stochasticity. Since $f_{\theta^*}$ is pre-trained on discrete graphs, feeding dense/weighted $\mathbf{A}_\phi$ during adaptation can create a distribution shift in message passing and lead to unstable optimization in low-label regimes.

Motivated by this gap, our DiP-G enforces discreteness throughout adaptation, the prompted structure $\mathbf{A}_{\text{prompt}} \in \{0, 1\}^{N \times N}$ is used consistently in both training and inference, aligning the adaptation input domain with the pre-training domain of $f_{\theta^*}$.

## 4. Method: DiP-G

We present **DiP-G** (Discrete Prompting for Graphs), a discrete graph prompting framework for few-shot graph adaptation, as shown in Figure 2. DiP-G learns task-specific prompted structures directly in the combinatorial space and avoids continuous relaxations. Our approach combines (i) a local, parametric prompting module for scalable candidate scoring, (ii) a perturbed Top-$k$ solver that produces hard $k$-sparse discrete prompts, and (iii) an implicit-gradient estimator with an adaptive screening routine to reduce the cost of the backward pass.

### 4.1. Parametric Local Subgraph Prompting

Optimizing a prompt over global adjacency matrix incurs $O(N^2)$ complexity and is impractical for large-scale graphs. Therefore, we parameterize the prompts locally. For each anchor node $i \in \mathcal{V}$, we define a candidate neighbor set $\mathcal{N}_{\text{cand}}(i)$, and restrict prompt decisions to edges between $i$ and $\mathcal{N}_{\text{cand}}(i)$.

We process anchors in mini-batches. Let $\mathcal{B} \subseteq \mathcal{V}$ be a batch of size $B$, then we construct a dense candidate index tensor $\mathbf{C} \in \mathbb{N}^{B \times m}$, where $m$ is a fixed candidate budget. To keep the Top-$k$ solver in Eq. (5) meaningful when $|\mathcal{N}_{\text{cand}}(i)| < k$, we pad with a a dummy candidate implemented as a self-loop, so selecting the dummy does not introduce an additional effective edge beyond the standard self-loop convention in GNN message passing.

To facilitate the learning of structural weights, we require node representations. To avoid a circular dependency in which the prompted structure changes the features used to generate itself, we compute and cache node embeddings only once using the frozen encoder on the initial structure

$$\mathbf{H}^{(0)} = f_{\theta^*}(\mathbf{A}_{\text{init}}, \mathbf{X}), \quad \mathbf{h}_i := \mathbf{H}_i^{(0)}. \qquad (3)$$

This caching mechanism not only ensures stability but also significantly accelerates the training loop.

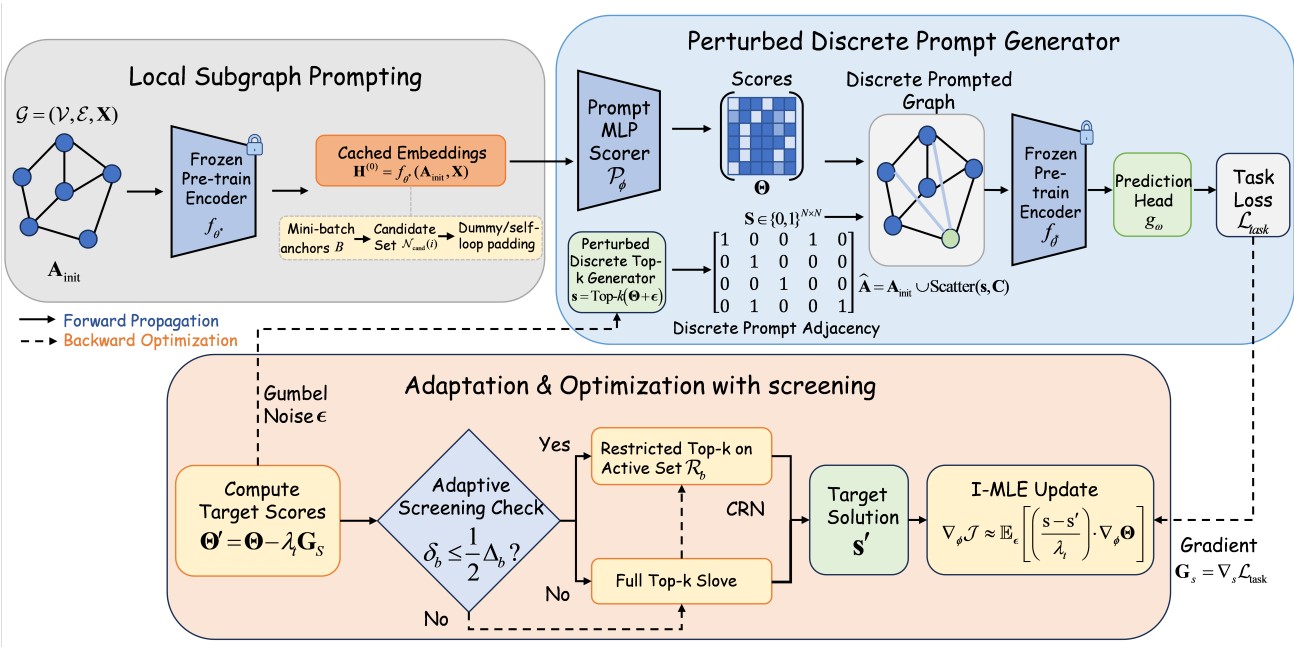

*Figure 2.* The framework of DiP-G. We score local candidates with a prompt module, generate hard $k$-sparse discrete prompts via a perturbed Top-$k$ solver, and optimize prompts using an I-MLE estimator with screening for efficient training.

We parameterize the edge scoring function $\mathcal{P}_\phi$ with an MLP. For an anchor $u$ and a candidate $v$, the unnormalized score $\Theta_{uv}$ is

$$\Theta_{uv} = \mathcal{P}_\phi(\mathbf{h}_u, \mathbf{h}_v) = \mathbf{w}_2^\top \sigma\left(\mathbf{W}_1(\mathbf{h}_u \,\|\, \mathbf{h}_v) + \mathbf{b}_1\right) + b_2, \tag{4}$$

where $\|$ denotes concatenation and $\phi = \{\mathbf{W}_1, \mathbf{b}_1, \mathbf{w}_2, b_2\}$. Applying $\mathcal{P}_\phi$ to the batch tensor $\mathbf{C}$, we obtain a score matrix $\Theta \in \mathbb{R}^{B \times m}$. Entries corresponding to dummy/self-loop candidates are fixed to value like $0$ to provide a sparsity bias, rather than being masked to $-\infty$.

### 4.2. Perturbed Discrete Prompt Generator

We generate discrete graph prompts using a perturbed combinatorial solver, replacing standard continuous Softmax-style relaxations. For each anchor, prompt construction is simplified to select exactly $k$ candidates under a hard cardinality constraint.

Let $\boldsymbol{\epsilon} \in \mathbb{R}^{B \times m}$ be i.i.d. Gumbel noise with entries $\epsilon_{bj} \sim$ Gumbel$(0, 1)$. For the $b$-th anchor in the batch, we define a binary selection vector $\mathbf{s}_b \in \{0, 1\}^m$ as the solution of

$$\mathbf{s}_b(\Theta_b, \boldsymbol{\epsilon}_b) = \arg\max_{\mathbf{z} \in \{0,1\}^m} \langle \mathbf{z}, \Theta_b + \boldsymbol{\epsilon}_b \rangle \quad \text{s.t.} \quad \|\mathbf{z}\|_0 = k. \tag{5}$$

This problem gets a closed-form solution via Top-$k$ on the perturbed scores. Let $\mathcal{I}_k(\mathbf{v})$ denote the indices of the $k$ largest entries of $\mathbf{v}$, so

$$[\mathbf{s}_b]_j = \mathbb{I}(j \in \mathcal{I}_k(\Theta_b + \boldsymbol{\epsilon}_b)). \tag{6}$$

This operation ensures that exactly $k$ candidates are selected for each node. When padded dummy entries are selected, they do not introduce effective edges beyond the standard self-loop convention, while preserving strict $k$-feasibility of the solver.

We then scatter the local selections back to the global prompt adjacency $\mathbf{S} \in \{0, 1\}^{N \times N}$ using the candidate index tensor $\mathbf{C}$, and form the prompt graph as

$$\widetilde{\mathbf{A}} = \mathbf{A}_{\text{init}} \cup \text{Scatter}(\{\mathbf{s}_b\}_{b=1}^B, \mathbf{C}). \tag{7}$$

In implementation, the union is realized by $\widetilde{\mathbf{A}} = \min(\mathbf{A}_{\text{init}} + \mathbf{S}, 1)$ to ensure binary entries. For gradient computation, we treat the message-passing adjacency on the candidate support as real-valued edge weights, while the prompt generation remains discrete. Unlike relaxation-based prompting, $\widetilde{\mathbf{A}}$ is discrete in both training and inference.

### 4.3. Optimization via Implicit Differentiation

The discrete solver in Eq. (6) makes the mapping $\Theta \mapsto \mathbf{S}$ piecewise constant, so standard backpropagation is not applicable. We therefore optimize the prompt parameters $\phi$ using I-MLE (Niepert et al., 2021), which provides a finite-difference estimator aimed at combinatorial MAP solvers.

We define the **training objective** as the expected task loss under the distribution of discrete prompted graphs induced

by perturbation noise $\epsilon \sim \mathcal{D}_\epsilon$:

$$\mathcal{J}(\phi) = \mathbb{E}_{\epsilon \sim \mathcal{D}_\epsilon} \left[ \mathcal{L}_{task} \left( g_\omega(f_{\theta^*}(\widetilde{\mathbf{A}}(\mathbf{\Theta}(\phi), \epsilon), \mathbf{X})) \right) \right]. \tag{8}$$

Let $\mathbf{G}_S = \nabla_{\mathbf{S}} \mathcal{L}_{task}$ be the gradient of the task loss with respect to the prompt adjacency $\mathbf{S}$, treating $\mathbf{S}$ as message-passing edge weights, and we compute $\mathbf{G}_S$ only on the candidate support. I-MLE forms a target score tensor $\mathbf{\Theta}' = \mathbf{\Theta} - \lambda_t \mathbf{G}_S$ and estimates the gradient of the expected objective by comparing the discrete solutions under $\mathbf{\Theta}$ and $\mathbf{\Theta}'$:

$$\nabla_\phi \mathcal{J}(\phi) \approx \mathbb{E}_\epsilon \left[ \left( \frac{\mathbf{S}(\mathbf{\Theta}, \epsilon) - \mathbf{S}(\mathbf{\Theta}', \epsilon)}{\lambda_t} \right) \cdot \nabla_\phi \mathbf{\Theta} \right]. \tag{9}$$

We use Common Random Numbers (CRN) by reusing the same noise realization $\epsilon$ for both solves, which couples $\mathbf{S}(\mathbf{\Theta}, \epsilon)$ and $\mathbf{S}(\mathbf{\Theta}', \epsilon)$ and reduces the variance of the difference estimator.

The step size $\lambda_t$ controls the strength of the target perturbation and then determines the bias variance trade-off relationship of gradient estimation. To accommodate changing gradient during training, we adapt $\lambda_t$ using a robust scale estimate of $|\mathbf{G}_S|$. Let $q$ be a fixed quantile, and update an exponential moving average $s_t$ as follows:

$$s_t = \beta s_{t-1} + (1 - \beta) \cdot \text{Quantile}(|\mathbf{G}_S|, q). \tag{10}$$

We then set

$$\lambda_t = \text{clip} \left( \frac{\lambda_0}{s_t + \xi}, \lambda_{\min}, \lambda_{\max} \right). \tag{11}$$

This keeps the perturbation $\lambda_t \mathbf{G}_S$ effective yet bounded, preventing the target solution from drifting extremely far from the current solution while avoiding manual tuning of $\lambda$.

### 4.4. Screened Target Solve

Eq. (9) requires two Top-$k$ solves per step: a forward solve for $\mathbf{s}_b$ under $\mathbf{\Theta}_b$ and a target solve for $\mathbf{s}'_b$ under $\mathbf{\Theta}'_b$. Running the full Top-$k$ on $m$ candidates twice can be costly. We therefore introduce an adaptive active-set screening rule that restricts the target solve to a small candidate subset when it is guaranteed to be exact.

During the forward solve, we compute not only the top-$k$ indices but also a buffered set of top-$(k + r)$ indices, where $r$ is a small constant. Let $\mathcal{R}_b$ denote these indices for anchor $b$, and let $v_{b,(j)}$ be the $j$-th largest entry of the perturbed score vector $\mathbf{\Theta}_b + \epsilon_b$. We define the screening gap

$$\Delta_b = v_{b,(k)} - v_{b,(k+r+1)}. \tag{12}$$

For the target solve, the score perturbation is $\mathbf{\Theta}'_b - \mathbf{\Theta}_b = -\lambda_t[\mathbf{G}_S]_b$. We summarize its magnitude by

$$\delta_b = \|\lambda_t[\mathbf{G}_S]_b\|_\infty. \tag{13}$$

We compute $\mathbf{s}'_b$ using a restricted Top-$k$ on $\mathcal{R}_b$ whenever the perturbation cannot change the membership of the top-$k$ set beyond the buffer:

$$\mathbf{s}'_b = \begin{cases} \text{Top-}k\left( (\mathbf{\Theta}'_b + \epsilon_b)|_{\mathcal{R}_b} \right) & \text{if } \delta_b \leq \frac{1}{2}\Delta_b, \\ \text{Top-}k(\mathbf{\Theta}'_b + \epsilon_b) & \text{otherwise.} \end{cases} \tag{14}$$

When the condition holds, the restricted solve is exact while reducing the sorting cost from $O(m \log k)$ to $O((k + r) \log k)$. Since typically $k + r \ll m$, this substantially accelerates the target solve in practice, with a fallback to the full solve when the condition is not met.

## 5. Analysis of DiP-G

In this section, we present a comprehensive analysis of our proposed framework. The overall algorithm of DiP-G can be found in Appendix B.

### 5.1. Complexity Analysis

We analyze the per-iteration overhead introduced by DiP-G under the mini-batch local prompting setting. For a mini-batch of $B$ anchors, we score a fixed candidate budget of $m$ nodes per anchor using the prompt MLP on cached embeddings, which costs $O(Bmd^2)$ for a two-layer MLP. The discrete prompt is obtained by a perturbed Top-$k$ selection, which can be implemented by partial sorting with cost $O(B \, m \log k)$ for the forward solve.

The I-MLE update in Eq. (9) requires an additional Top-$k$ solve under $\mathbf{\Theta}'$. So our screening rule restricts this target solve to a buffered active set of size $k + r$ when possible, reducing the typical cost from $O(B \, m \log k)$ to $O(B \, (k + r) \log k)$. Overall, the expected backward cost can be summarized as

$$\mathbb{E}[T_{\text{bwd}}] = \sum_{b=1}^{B} \left( T(k + r) + \mathbb{P}(\text{Fallback}_b) \cdot T(m) \right), \tag{15}$$

where $T(n)$ denotes the cost of Top-$k$ over $n$ candidates. Since $k + r \ll m$ typically, screening substantially reduces the overhead of the target solve when fallback events are rare.

### 5.2. Theoretical Analysis

In this section, we provide a theoretical justification for the key components of DiP-G. First, the perturbed Top-$k$ solver enforces the structural constraint by construction that for every anchor $b$, the selected indicator vector satisfies $\|\mathbf{s}_b\|_0 = k$. Furthermore dummy/self-loop padding ensures feasibility when a neighborhood contains fewer than $k$ valid candidates.

Our main theoretical result concerns the screening rule used to accelerate the target solve in the I-MLE estimator. During

*Table 1.* 5-shot node classification accuracy (%) on five benchmark datasets under four pre-training strategies. Best in **bold** and runner-up underlined.

| Pre-training | Method | Cora | PubMed | Amazon | Minesweeper | Flickr |
|---|---|---|---|---|---|---|
| **GraphCL** | Frozen+Linear | 55.69±5.74 | 67.30±6.26 | 23.19±7.21 | 67.59±6.30 | 29.31±8.91 |
| | GPPT | 61.50±4.49 | 65.75±3.99 | 24.27±3.74 | 65.44±8.97 | 24.64±3.15 |
| | All-in-one | 52.33±4.55 | 65.78±8.65 | 22.82±6.09 | 63.82±8.63 | 21.57±4.64 |
| | GraphPrompt | 62.12±3.28 | 67.01±4.56 | 21.71±2.93 | 61.19±3.50 | 21.92±3.72 |
| | GraphPrompt+ | 58.91±3.12 | 66.26±5.75 | 23.83±2.30 | 61.64±6.36 | 24.43±4.62 |
| | ProNoG | 60.01±7.03 | 68.17±4.82 | 23.26±2.42 | 65.48±3.40 | 26.17±5.18 |
| | EdgePrompt | 58.60±4.46 | 67.76±3.01 | 25.81±2.86 | 66.71±4.74 | 24.83±2.78 |
| | EdgePrompt+ | 62.88±6.43 | 67.41±5.25 | 26.88±4.07 | 68.18±5.26 | 25.57±3.04 |
| | GraphTOP | 63.44±4.21 | 68.28±4.15 | 27.43±7.02 | 68.25±7.14 | 30.93±9.07 |
| | **DiP-G (Ours)** | **65.12±2.15** | **70.44±1.88** | **29.23±3.41** | **70.84±2.12** | **32.50±3.22** |
| **SimGRACE** | Frozen+Linear | 40.68±2.29 | 54.59±6.02 | 24.58±4.18 | 60.58±6.42 | 26.78±5.29 |
| | GPPT | 44.83±4.67 | 52.25±5.91 | 24.27±3.74 | 59.62±4.80 | 22.11±3.56 |
| | All-in-one | 41.11±4.92 | 51.45±4.73 | 22.66±3.55 | 58.11±3.82 | 21.50±4.49 |
| | GraphPrompt | 47.02±3.87 | 55.74±5.80 | 21.24±2.78 | 58.72±4.37 | 19.72±4.54 |
| | GraphPrompt+ | 51.26±4.90 | 55.93±6.98 | 25.07±1.71 | 60.76±6.75 | 20.79±5.65 |
| | ProNoG | 42.44±2.97 | 55.11±5.98 | 22.53±2.65 | 63.03±2.74 | 25.44±4.45 |
| | EdgePrompt | 58.37±4.51 | 58.91±3.05 | 24.81±3.75 | 63.55±2.66 | 30.12±5.04 |
| | EdgePrompt+ | **62.40±7.97** | 61.04±3.25 | 27.18±3.03 | 64.18±3.71 | 28.50±4.08 |
| | GraphTOP | 57.76±3.83 | 58.64±5.42 | 25.67±3.34 | 61.25±5.08 | 27.70±5.69 |
| | **DiP-G (Ours)** | 58.43±2.02 | **61.92±2.61** | **28.18±2.13** | **65.22±2.45** | **31.12±3.18** |
| **DP-GPPT** | Frozen+Linear | 24.40±2.83 | 42.26±5.09 | 25.50±4.11 | 63.22±8.16 | 23.76±4.30 |
| | GPPT | 32.08±7.66 | 44.85±4.73 | 28.90±3.50 | 63.44±8.28 | 22.25±4.41 |
| | All-in-one | 26.67±6.24 | 41.11±4.92 | 24.49±3.51 | 59.97±4.67 | 18.09±4.30 |
| | GraphPrompt | 30.14±2.01 | 44.72±6.68 | 21.88±3.56 | 61.73±6.35 | 19.72±1.76 |
| | GraphPrompt+ | 33.42±2.91 | 45.17±6.91 | 24.34±1.72 | 61.15±3.37 | 21.02±4.22 |
| | ProNoG | 33.71±4.12 | 46.07±3.62 | 21.39±1.69 | 66.11±4.20 | 24.08±4.10 |
| | EdgePrompt | 36.34±3.25 | 46.15±3.66 | 29.81±3.57 | 67.71±3.23 | 29.56±4.32 |
| | EdgePrompt+ | **42.88±6.43** | 48.03±4.25 | 31.56±5.24 | 69.32±3.26 | **30.03±4.77** |
| | GraphTOP | 33.97±2.43 | 46.52±6.18 | 32.41±7.18 | 66.67±3.83 | 25.95±3.17 |
| | **DiP-G (Ours)** | 37.52±1.95 | **49.17±2.42** | **35.23±3.19** | **70.35±2.22** | 29.88±2.56 |
| **DP-GraphPrompt** | Frozen+Linear | 50.15±5.88 | 66.26±5.69 | 25.00±6.78 | 65.90±7.36 | 23.75±3.26 |
| | GPPT | 52.13±7.15 | 63.16±8.25 | 25.38±5.78 | 62.53±8.91 | 24.16±3.88 |
| | All-in-one | 49.42±2.70 | 64.73±6.46 | 21.37±3.65 | 58.17±4.63 | 22.10±2.92 |
| | GraphPrompt | 52.35±4.82 | 68.16±8.23 | 22.76±2.81 | 58.01±3.26 | 21.15±1.46 |
| | GraphPrompt+ | 52.19±5.22 | 62.19±6.70 | 24.44±1.81 | 61.78±3.92 | 21.48±4.09 |
| | ProNoG | 52.49±5.43 | 67.68±5.02 | 23.79±2.04 | 59.88±8.50 | 24.74±1.10 |
| | EdgePrompt | 55.76±4.14 | 68.03±4.53 | 29.02±4.91 | 70.13±5.81 | 29.81±3.59 |
| | EdgePrompt+ | 57.14±3.38 | 69.12±3.77 | 29.56±5.33 | **74.03±5.11** | **32.09±4.93** |
| | GraphTOP | 56.44±4.72 | 68.14±5.47 | 29.74±7.72 | 69.90±6.46 | 30.43±3.52 |
| | **DiP-G (Ours)** | **57.29±2.18** | **70.03±2.33** | **31.13±2.54** | 71.52±2.65 | 30.33±2.90 |

forward solve, we store a buffered active set $\mathcal{R}_b$ containing the top-$(k+r)$ indices of the perturbed score vector $\Theta_b + \epsilon_b$. Let $v_{b,(j)}$ be the $j$-th largest entry of $\Theta_b + \epsilon_b$ and define the screening gap

$$\Delta_b = v_{b,(k)} - v_{b,(k+r+1)}. \qquad (16)$$

For the target solve, the score perturbation is $\Theta'_b - \Theta_b = -\lambda_t[\mathbf{G}_S]_b$; we measure its magnitude by $\delta_b = \|\lambda_t[\mathbf{G}_S]_b\|_\infty$.

**Theorem 5.1** (Exactness of the Screened Target Solve). *If* $\delta_b \leq \frac{1}{2}\Delta_b$, *then the target Top-$k$ index set satisfies*

$$\mathcal{I}_k(\Theta'_b + \epsilon_b) \subseteq \mathcal{R}_b, \qquad (17)$$

*and the restricted solve Top-$k$$((\Theta'_b + \epsilon_b)|_{\mathcal{R}_b})$ returns the same $\mathbf{s}'_b$ as the full solve Top-$k$$(\Theta'_b + \epsilon_b)$.*

**Proof.** See Appendix A.2.

Finally, DiP-G adopts I-MLE to optimize discrete prompts. Under standard regularity conditions (stated in Appendix A.1), the resulting finite-difference estimator is asymptotically consistent as $\lambda_t \to 0$, with an $O(\lambda_t)$ bias term for finite $\lambda_t$; we defer the detailed derivations to Appendix A.3.

## 6. Experiments

### 6.1. Experimental Setup

**Datasets.** We adopt five real-world graph datasets from diverse domains to evaluate few-shot node classification, including Cora (Yang et al., 2016),

*Table 2.* Few-shot node classification accuracy (%) under the 10-shot setting. **Bold** denotes the best result, and underlined indicates the second best.

| Pre-training | Method | Cora | CiteSeer | PubMed | ogbn-arxiv | Flickr |
|---|---|---|---|---|---|---|
| **GraphCL** | Frozen+Linear | 65.38±2.31 | 44.07±4.23 | 68.13±1.12 | 26.83±1.57 | 30.22±2.41 |
| | GPPT | 58.47±3.32 | 44.52±8.18 | 67.63±6.21 | 26.61±3.52 | 28.76±5.78 |
| | GraphPrompt | 63.62±2.54 | 46.27±2.96 | 67.81±1.89 | 25.57±1.11 | 26.82±1.84 |
| | All-in-one | 51.66±6.88 | 43.24±2.86 | 61.27±2.74 | 21.91±2.34 | 24.56±3.64 |
| | GPF | 70.14±1.96 | 47.46±4.12 | 70.67±1.34 | 27.52±1.46 | 27.64±2.16 |
| | GraphTOP | 65.41±2.06 | 44.12±3.84 | 68.24±1.06 | 26.79±1.24 | 29.86±1.49 |
| | EdgePrompt | 70.27±1.84 | 47.96±4.04 | 70.62±1.61 | 27.58±1.16 | 28.67±2.44 |
| | EdgePrompt+ | 74.16±3.31 | 53.12±4.14 | 72.74±2.46 | 28.84±1.16 | 30.81±2.24 |
| | **DiP-G (Ours)** | **75.62±1.45** | **55.23±2.85** | **74.39±1.50** | **30.42±1.21** | **33.54±1.87** |
| **SimGRACE** | Frozen+Linear | 62.27±3.04 | 45.56±3.64 | 60.66±1.94 | 27.14±0.96 | 30.41±1.84 |
| | GPPT | 60.16±5.04 | 40.36±7.04 | 62.24±6.19 | 27.21±3.34 | 30.26±6.24 |
| | GraphPrompt | 59.34±2.16 | 47.31±3.24 | 62.61±1.74 | 25.76±0.86 | 30.19±1.16 |
| | All-in-one | 49.91±2.84 | 43.86±2.94 | 60.11±2.06 | 20.14±2.96 | 29.57±3.64 |
| | GPF | 67.82±3.94 | 49.16±3.24 | 63.66±1.69 | 27.96±0.94 | 32.86±3.84 |
| | GraphTOP | 62.31±3.24 | 45.49±4.06 | 60.76±1.79 | 27.16±0.86 | 33.96±3.24 |
| | EdgePrompt | 68.34±3.96 | 49.41±3.34 | 63.76±1.59 | 27.94±1.06 | 33.64±3.51 |
| | EdgePrompt+ | **72.66±3.44** | 52.86±3.14 | 69.61±2.49 | 28.76±0.96 | 32.27±2.79 |
| | **DiP-G (Ours)** | 71.81±2.24 | **54.48±2.13** | **71.21±1.66** | **30.19±0.93** | **35.64±2.42** |
| **DP-GPPT** | Frozen+Linear | 34.21±3.14 | 28.49±3.26 | 45.16±4.04 | 16.06±1.74 | 31.86±5.34 |
| | GPPT | 48.56±6.04 | 36.06±5.94 | 56.64±9.29 | 23.66±1.79 | 29.66±6.69 |
| | GraphPrompt | 35.14±1.51 | 28.27±1.59 | 48.81±5.19 | 13.46±1.79 | 29.14±3.46 |
| | All-in-one | 35.21±1.64 | 27.26±2.54 | 47.16±1.59 | 16.66±0.41 | 32.36±2.34 |
| | GPF | 49.72±0.46 | 35.26±2.39 | 50.61±2.69 | 22.56±2.14 | 31.56±5.44 |
| | GraphTOP | 33.71±2.24 | 28.24±3.26 | 45.26±4.59 | 16.14±1.81 | 30.76±7.49 |
| | EdgePrompt | 50.49±0.86 | 34.66±2.94 | 50.96±2.44 | 22.66±2.14 | 30.84±6.51 |
| | EdgePrompt+ | 69.56±6.34 | 50.86±2.74 | 60.96±4.24 | 21.76±2.04 | 30.86±5.64 |
| | **DiP-G (Ours)** | **70.53±2.49** | **52.37±2.15** | **62.81±2.80** | **24.19±1.54** | **33.11±3.14** |
| **DP-GraphPrompt** | Frozen+Linear | 68.24±3.16 | 47.86±3.49 | 75.56±1.69 | 36.76±0.86 | 31.46±7.98 |
| | GPPT | 69.06±4.44 | 48.91±8.34 | 74.86±6.69 | 25.71±3.49 | 32.89±3.06 |
| | GraphPrompt | 69.04±2.51 | 50.36±2.14 | 75.81±1.34 | 36.91±0.81 | 30.46±5.24 |
| | All-in-one | 57.81±3.14 | 46.26±5.64 | 74.36±2.99 | 22.89±2.56 | 30.66±5.19 |
| | GPF | 72.36±2.84 | 51.16±3.64 | **77.86±2.34** | 36.96±1.09 | 29.81±8.84 |
| | GraphTOP | 68.39±3.66 | 48.41±3.54 | 75.66±1.69 | 36.69±1.11 | 29.46±8.19 |
| | EdgePrompt | 72.31±2.39 | 51.46±3.54 | 77.41±2.49 | 37.21±1.19 | 32.11±4.54 |
| | EdgePrompt+ | 75.16±3.04 | 56.16±2.59 | 76.76±2.04 | 37.36±1.44 | 34.56±7.04 |
| | **DiP-G (Ours)** | **76.43±1.89** | **57.62±1.91** | 77.58±1.54 | **38.52±1.15** | **36.20±3.54** |

PubMed (Yang et al., 2016), Amazon (Platonov et al., 2023), Minesweeper (Platonov et al., 2023), and Flickr (Zeng et al., 2019). Detailed statistics and information are provided in Appendix D.1.

**Pre-training strategies.** To evaluate compatibility with different pre-training objectives, we consider four representative strategies. For contrastive pre-training, we use GraphCL (Hafidi et al., 2020) and SimGRACE (Xia et al., 2022). For generation-based pre-training, we follow GPPT (Sun et al., 2022) and GraphPrompt (Liu et al., 2023) to pre-train GNN encoders via link prediction, denoted as DP-GPPT and DP-GraphPrompt, respectively. Further details of these pre-training pipelines are deferred to Appendix D.2.

**Baselines.** We compare DiP-G with strong prompting baselines that cover both feature-level and edge-level designs, including GPPT (Sun et al., 2022), All-in-One (Sun

et al., 2023), GraphPrompt, GraphPrompt+ (Liu et al., 2023), GPF (Fang et al., 2023), ProNoG (Yu et al., 2025), GraphTOP (Fu et al., 2025b), as well as EdgePrompt and EdgePrompt+ (Fu et al., 2025a). We additionally adopt Frozen+Linear, which keeps the pre-trained encoder frozen and trains only a linear classifier on its node representations. All methods use the same frozen backbone and the same few-shot protocol for a fair comparison, and specific settings follow their original papers whenever applicable. More information on these baselines can be found in Appendix D.3.

**Implementation details.** We use a 2-layer GCN (Kipf, 2016) with hidden dimension 128 as the frozen encoder for all methods. The results in table 1 are reported under the 5-shot setting. We train with Adam (Kingma, 2014) using learning rate $5 \times 10^{-3}$ for 500 epochs and report mean ± std over 10 random seeds. We construct candidates from 2-hop neighborhoods and set the anchor batch size to $B = 32$ for all datasets. We set the candidate budget $m = 128$, sparsity

*Table 3.* 5-shot node classification results under GraphCL and DP-GraphPrompt on ogbn-arxiv.

| Method | GraphCL | LP-GraphPrompt |
|---|---|---|
| GPPT | 18.45±1.83 | 27.51±1.85 |
| All-in-one | 17.85±3.22 | 16.43±4.02 |
| GraphPrompt | 21.29±2.53 | 32.89±1.84 |
| GraphPrompt+ | 21.86±2.91 | 31.56±1.24 |
| ProNoG | 20.60±3.42 | 32.25±2.57 |
| EdgePrompt | 21.95±1.68 | 32.72±1.80 |
| EdgePrompt+ | 23.25±1.22 | 31.50±1.85 |
| GraphTOP | 23.54±2.29 | 33.86±2.81 |
| **DiP-G** | **24.84±1.48** | **35.21±1.64** |

$k = 8$, and screening buffer $r = 4$ by default. We do not tune and per dataset, and we do not rely on a large held-out validation set. For the I-MLE estimator, we follow the default configuration in (Niepert et al., 2021) and do not tune these hyperparameters unless explicitly stated.

## 6.2. Performance Comparison

Table 1 and 2 compares DiP-G with strong feature-prompting and topology-prompting baselines across five datasets and four pre-training strategies. DiP-G delivers consistently strong performance, and attains the best results under GraphCL pre-training across all datasets. For SimGRACE and link-prediction pre-training like DP-GPPT and DP-GraphPrompt, DiP-G remains among the top-performing methods and frequently achieves the best performance. Additionally, we observe that EdgePrompt+ and GraphTOP are strong baselines compared to other graph prompting methods. They perform well especially in Flickr under DP-GPPT and DP-GraphPrompt. Across settings, DiP-G improves over feature-side prompting approaches in a more reliable manner, suggesting that directly editing the message-passing structure provides a stronger adaptation signal in the low-label regime. Moreover, compared with relaxation-based topology prompting, DiP-G exhibits more stable behavior across pre-training objectives, supporting our motivation that enforcing discrete prompts throughout adaptation mitigates the discretization gap. We also conduct comparison on 5-shot node classification on ogbn-arxiv on table 3 and 100-shot graph classification in Appendix C.

## 6.3. Parameter Sensitivity

Figure 3 and Figure 4 studies the sensitivity of DiP-G to the sparsity level $k$, the candidate budget $m$ and the buffer size $r$ under GraphCL pre-training on Cora and PubMed.

**Effect of sparsity** $k$. As $k$ increases from small values, accuracy improves on both datasets, indicating that selecting too few prompt edges limits the capacity of structural

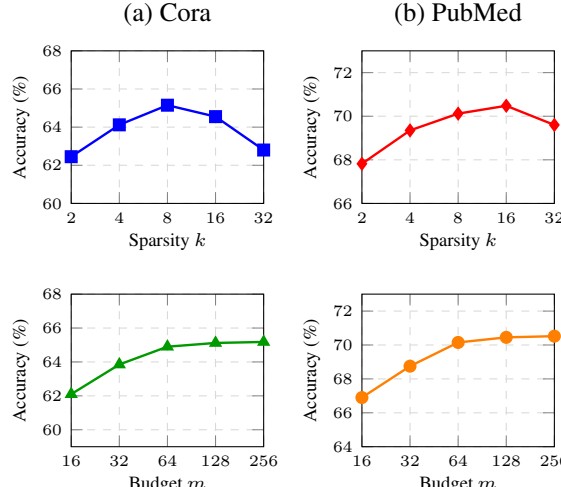

*Figure 3.* Parameter sensitivity to $k$ and $m$ of GraphCL on Cora and PubMed.

adaptation. However, further increasing $k$ does not lead to consistent gains and can slightly degrade performance, as overly dense prompts may introduce noisy connections and weaken the benefit of a targeted structural edit in the few-shot regime. Across the range, the best performance is achieved with moderate sparsity $k = 8$, which provides a stable trade-off between expressivity and robustness.

**Effect of budget** $m$. Increasing the candidate budget consistently improves accuracy when $m$ is small, as a larger candidate pool allows the Top-$k$ solver to identify more informative prompt edges. Once $m$ reaches a moderate scale, the improvements saturate, indicating that the local candidate set already covers the most useful structural alternatives, and additional candidates mainly increase the computation. This trend supports our default choice $m = 128$, which achieves strong performance while keeping the per-iteration overhead manageable.

**Effect of screening buffer size** $r$. Figure 4 reports the fallback rate of our screened target solve as a function of the buffer size $r$. We observe a sharp drop in fallback when $r$ increases from small values, followed by a clear saturation regime where fallback becomes rare on both Cora and PubMed. This trend is consistent with our screening condition that enlarging the buffered active set increases the score margin between the $k$-th selected candidate and candidates outside the buffer, making it less likely that the target perturbation changes the Top-$k$ membership beyond the buffer. In practice, moderate buffers are suitable for screened target solves, while further enlarging $r$ brings limited additional benefit but increases the size of the restricted candidate set in the target solve.

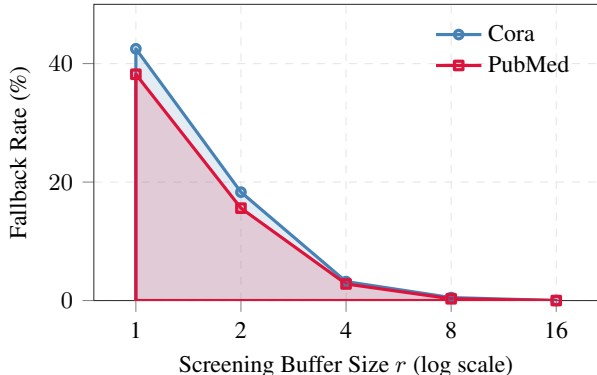

*Figure 4.* Impact of screening buffer size $r$ on fallback rate.

*Table 4.* Ablation study of DiP-G.

|  | Cora | PubMed | Amazon | Mine. |
|---|---|---|---|---|
| Frozen+Linear | 55.69±5.74 | 67.33±6.26 | 23.19±7.21 | 67.59±6.32 |
| Soft prompt | 61.54±3.12 | 67.84±2.55 | 25.47±3.62 | 68.10±4.28 |
| Hard prompt + STE | 63.84±2.49 | 69.13±2.12 | 27.80±2.98 | 69.53±2.86 |
| **DiP-G (Full)** | **65.12±2.15** | **70.45±1.88** | **29.25±3.48** | **70.84±2.11** |

### 6.4. Ablation Study

We ablate the key components of DiP-G under the same GraphCL pre-training, 5-shot protocol, and training budget as the main experiments. Table 4 reports results on four datasets. We include Frozen+Linear as a reference, which is the same baseline already reported in Table 1. It provides a clear lower bound when no prompting. Compared to this baseline, all prompt-based variants improve performance, showing that structural editing is helpful for few-shot adaptation. We then replace our perturbed Top-$k$ generator with a soft prompt variant that learns continuous edge weights during training and applies a Top-$k$ discretization at test time to keep the same sparsity constraint. This relaxed training consistently underperforms the full model, which matches our motivation that the discretization gap can hurt adaptation. Finally, we keep the hard Top-$k$ prompt in the forward pass but replace I-MLE with a straight-through estimator. This variant is stronger than the soft relaxation but still worse than DiP-G, suggesting that I-MLE provides a better learning signal for the combinatorial prompt generator.

### 6.5. Efficiency Analysis

We evaluate the training efficiency of DiP-G on Cora and Flickr, and compare it with a representative feature-prompt baseline (GPPT) and a topology-prompt baseline (Graph-TOP). Figure 5 reports the per-epoch training time and peak GPU memory. As we can see, DiP-G is consistently more efficient than GraphTOP while remaining close to GPPT. On both datasets, DiP-G requires much less GPU memory than GraphTOP. This is expected since DiP-G operates on local candidate sets and uses cached backbone embeddings,

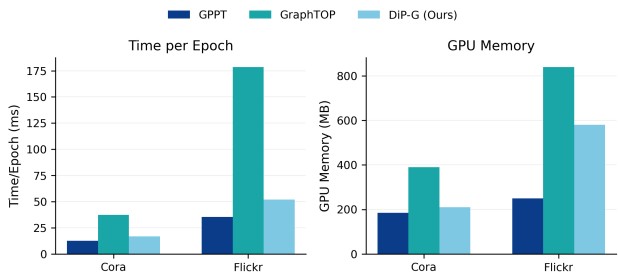

*Figure 5.* Training efficiency comparison on Cora and Flickr (time per epoch and peak GPU memory).

avoiding heavy global structure operations. At the same time, DiP-G uses more memory than GPPT because it maintains candidate scores and performs discrete Top-$k$ selection during training.

In terms of runtime, DiP-G is also substantially faster than GraphTOP on both datasets, especially on Flickr. The additional overhead compared to GPPT comes from the discrete prompt generator and the second target solve in the I-MLE update, while our screening rule reduces this cost in most iterations. Figure 5 suggest that DiP-G offers a better efficiency performance for discrete topology prompting, and scales more favorably to larger graphs.

## 7. Conclusion

We proposed DiP-G, a discrete graph prompting framework for few-shot adaptation that learns hard $k$-sparse topology prompts directly in the combinatorial space. By keeping the prompted graph discrete throughout training and inference, DiP-G avoids the discretization gap that affects relaxed prompting methods. Our approach combines local candidate scoring, a perturbed Top-$k$ prompt generator, and I-MLE optimization with an adaptive screening rule for efficient training. Experiments across diverse datasets and pre-training strategies show that DiP-G achieves strong and stable few-shot performance, while maintaining a favorable efficiency performance. In future work, we plan to extend DiP-G to broader graph tasks and more adaptive candidate construction to further improve scalability and generalization.

## Impact Statement

This paper presents work to advance the field of machine learning, by improving few-shot adaptation of pre-trained graph neural networks via discrete graph prompting. Our method is general-purpose and does not introduce new capabilities for data collection on surveillance. Potential social impacts depend on the downstream application of GNNs, and these applications can raise well-known concerns such as privacy.

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

# A. Deferred Proofs and Additional Derivations

## A.1. Regularity Assumptions

**Assumption A.1** (Regularity). (i) The noise distribution $\mathcal{D}_\epsilon$ is absolutely continuous with bounded density. (ii) For fixed features $\mathbf{X}$, the composed map

$$\Phi(\mathbf{S}) := \mathcal{L}_{\text{supp}}(g_\omega(f_{\theta^*}(\mathbf{A}_{\text{init}} \cup \mathbf{S}, \mathbf{X}))) \tag{18}$$

is $L_\Phi$-Lipschitz with respect to the $\ell_1$ norm on the candidate support, i.e.,

$$|\Phi(\mathbf{S}) - \Phi(\mathbf{S}')| \le L_\Phi \|\mathbf{S} - \mathbf{S}'\|_1. \tag{19}$$

**Discussion of Assumption A.1(ii).** This condition is standard in analyses of gradient estimators for discrete solvers. In practice, common message-passing GNN layers are Lipschitz with respect to edge-weight perturbations under mild stability conditions, e.g., normalized aggregation, Lipschitz nonlinearities, and bounded parameter norms. For instance, a GCN-type layer

$$\mathbf{H}^{(\ell+1)} = \sigma(\widehat{\mathbf{D}}^{-1/2}\widehat{\mathbf{A}}\widehat{\mathbf{D}}^{-1/2}\mathbf{H}^{(\ell)}\mathbf{W}) \tag{20}$$

is Lipschitz in $\widehat{\mathbf{A}}$ when $\sigma$ is Lipschitz and $\|\mathbf{W}\|_2$ is bounded; composing finitely many such layers with a Lipschitz loss yields an overall Lipschitz map $\Phi$. We use this assumption to state an $O(\lambda_t)$ bias bound for the I-MLE finite-difference estimator below.

## A.2. Proof of Theorem 5.1

*Proof.* Fix an anchor $b$ and abbreviate $\mathbf{v} = \mathbf{\Theta}_b + \boldsymbol{\epsilon}_b$ and $\mathbf{v}' = \mathbf{\Theta}'_b + \boldsymbol{\epsilon}_b = \mathbf{v} - \lambda_t[\mathbf{G}_S]_b$. Let $\mathcal{R}_b$ be the indices of the top-$(k+r)$ entries of $\mathbf{v}$, and define $\Delta_b = v_{(k)} - v_{(k+r+1)}$.

Consider any $u \notin \mathcal{R}_b$. Then $v_u \le v_{(k+r+1)}$. Also, for any $w$ in the top-$k$ set of $\mathbf{v}$ we have $v_w \ge v_{(k)}$. Therefore,

$$v_w - v_u \ge v_{(k)} - v_{(k+r+1)} = \Delta_b. \tag{21}$$

Under the target perturbation, each coordinate changes by at most $\delta_b = \|\lambda_t[\mathbf{G}_S]_b\|_\infty$, hence

$$v'_w - v'_u = (v_w - v_u) - \lambda_t([\mathbf{G}_S]_{b,w} - [\mathbf{G}_S]_{b,u}) \ge \Delta_b - 2\delta_b. \tag{22}$$

If $\delta_b \le \frac{1}{2}\Delta_b$, then $v'_w > v'_u$. Thus no index $u \notin \mathcal{R}_b$ can outrank any index $w$ that remains in the forward top-$k$ band under the target scores. In particular, all indices in the target top-$k$ set must lie within $\mathcal{R}_b$, which implies Eq. (**??**) and the exactness of the restricted solve. $\square$

## A.3. Bias Analysis for the I-MLE Finite-Difference Estimator

We analyze the bias of the I-MLE finite-difference estimator used in Eq. (9). Let

$$J(\mathbf{\Theta}) = \mathbb{E}_{\boldsymbol{\epsilon} \sim \mathcal{D}_\epsilon}\big[\Phi(\mathbf{S}(\mathbf{\Theta}, \boldsymbol{\epsilon}))\big], \tag{23}$$

where $\Phi$ is defined in Assumption A.1. The I-MLE construction compares the discrete solutions obtained from $\mathbf{\Theta}$ and $\mathbf{\Theta}' = \mathbf{\Theta} - \lambda\mathbf{G}_S$ under a shared noise realization.

A first-order Taylor expansion of the smoothed objective around $\mathbf{\Theta}$ gives

$$J(\mathbf{\Theta} - \lambda\mathbf{G}_S) = J(\mathbf{\Theta}) - \lambda\langle\nabla_{\mathbf{\Theta}} J(\mathbf{\Theta}), \mathbf{G}_S\rangle + O(\lambda^2). \tag{24}$$

Under Assumption A.1, results in (Niepert et al., 2021) relate the expected discrete difference quotient to the gradient of the smoothed objective. In particular, for a suitable choice of $\mathbf{\Theta}'$,

$$\mathbb{E}\left[\frac{\mathbf{S}(\mathbf{\Theta}, \boldsymbol{\epsilon}) - \mathbf{S}(\mathbf{\Theta}', \boldsymbol{\epsilon})}{\lambda}\right] = \nabla_{\mathbf{\Theta}} J(\mathbf{\Theta}) + O(\lambda). \tag{25}$$

The $O(\lambda)$ term arises from linearizing the decision boundary of the Top-$k$ map under perturbations. Consequently, for finite $\lambda_t$, the estimator exhibits a bias that is first-order in $\lambda_t$.

---

**Algorithm 1** Training DiP-G with Adaptive Screening

---

**Require:** Graph $\mathcal{G} = (\mathcal{V}, \mathcal{E}, \mathbf{X})$, frozen encoder $f_{\theta^*}$, support set $\mathcal{S}_{\text{supp}}$, hyperparams $k, m, r$.
**Ensure:** Optimized prompt parameters $\phi$ and prediction head $\omega$.
 1: **// Phase 1: Initialization & Caching**
 2: Compute and cache node embeddings on initial structure:
 3: $\mathbf{H}^{(0)} \leftarrow f_{\theta^*}(\mathbf{A}_{\text{init}}, \mathbf{X})$  {Eq. (3)}
 4: Initialize prompt MLP $\mathcal{P}_\phi$ and head $g_\omega$.
 5: **// Phase 2: Training Loop**
 6: **while** not converged **do**
 7:   Sample a mini-batch of anchors $\mathcal{B} \subseteq \mathcal{S}_{\text{supp}}$.
 8:   Retrieve candidate tensor $\mathbf{C} \in \mathbb{N}^{B \times m}$ for $\mathcal{B}$.
 9:   Compute scores $\mathbf{\Theta} \leftarrow \mathcal{P}_\phi(\mathbf{H}^{(0)}[\mathcal{B}], \mathbf{H}^{(0)}[\mathbf{C}])$.   {Eq. (4)}
10:   Sample Gumbel noise $\boldsymbol{\epsilon} \sim \text{Gumbel}(0, 1)$.
11:   **// Forward: Perturbed Discrete Solve**
12:   Get Top-$k$ indices $\mathcal{I}_k$ and buffered Top-$(k + r)$ indices $\mathcal{R}$ based on $(\mathbf{\Theta} + \boldsymbol{\epsilon})$.
13:   Construct binary prompt vectors $\mathbf{s} \in \{0, 1\}^{B \times m}$ using $\mathcal{I}_k$.
14:   Form discrete graph $\widetilde{\mathbf{A}} \leftarrow \mathbf{A}_{\text{init}} \cup \text{Scatter}(\mathbf{s}, \mathbf{C})$.
15:   Prediction $\hat{\mathbf{Y}} \leftarrow g_\omega(f_{\theta^*}(\widetilde{\mathbf{A}}, \mathbf{X}))$.
16:   Compute task loss $\mathcal{L} \leftarrow \mathcal{L}_{\text{task}}(\hat{\mathbf{Y}}, \mathbf{Y}_\mathcal{B})$.
17:   **// Backward: Gradient Estimation with Screening**
18:   Compute structural gradient $\mathbf{G}_S \leftarrow \nabla_{\mathbf{S}} \mathcal{L}$ (treating $\mathbf{s}$ as continuous).
19:   Update step size $\lambda_t$ adaptively via Eq. (11).
20:   Compute target scores $\mathbf{\Theta}' \leftarrow \mathbf{\Theta} - \lambda_t \mathbf{G}_S$.
21:   **for** each anchor $b \in \mathcal{B}$ **do**
22:     Calculate gap $\Delta_b$ and perturbation magnitude $\delta_b$.
23:     **if** $\delta_b \leq \frac{1}{2}\Delta_b$ **then**
24:       $\mathbf{s}'_b \leftarrow \text{Top-}k((\mathbf{\Theta}'_b + \boldsymbol{\epsilon}_b)|_{\mathcal{R}_b})$   {**Screened Solve**: Sort only $k + r$ items}
25:     **else**
26:       $\mathbf{s}'_b \leftarrow \text{Top-}k(\mathbf{\Theta}'_b + \boldsymbol{\epsilon}_b)$   {**Full Solve**: Sort $m$ items}
27:     **end if**
28:   **end for**
29:   Estimate gradient via I-MLE:
30:   $\nabla_\phi \mathcal{J} \approx \frac{1}{\lambda_t}(\mathbf{s} - \mathbf{s}') \cdot \nabla_\phi \mathbf{\Theta}$.
31:   Update $\phi$ and $\omega$ using Adam.
32: **end while**

---

## A.4. Mini-batch Variance Bound

We provide a simple bound for the variance reduction induced by averaging over $B$ independent anchors. Define the *structure-level* finite-difference term for anchor $b$ as

$$\widehat{\mathbf{g}}_b = \frac{\mathbf{s}_b - \mathbf{s}'_b}{\lambda_t} \in \mathbb{R}^m. \tag{26}$$

This term is the discrete difference quotient appearing in Eq. (9) before multiplication by $\nabla_\phi \mathbf{\Theta}$.

**Proposition A.2** (Mini-batch Variance Bound). *Assume the anchor-wise noises $\{\boldsymbol{\epsilon}_b\}_{b=1}^B$ are i.i.d. and the loss decomposes over anchors. Then*

$$\mathbb{E}\left\| \frac{1}{B} \sum_{b=1}^B \widehat{\mathbf{g}}_b - \mathbb{E}[\widehat{\mathbf{g}}_1] \right\|_2^2 \leq \frac{2k}{B\lambda_t^2}. \tag{27}$$

*Proof.* Let $\Delta\mathbf{s}_b = \mathbf{s}_b - \mathbf{s}'_b$. Since both $\mathbf{s}_b$ and $\mathbf{s}'_b$ are $k$-sparse binary vectors, their supports can differ on at most $2k$ indices,

*Table 5.* Accuracy (%) on 100-shot graph classification tasks. Best in **bold** and runner-up underlined.

| Pre-training | Method | DD | NCI1 | NCI109 | Mutagenicity |
|---|---|---|---|---|---|
| **GraphCL** | Frozen+Linear | 62.86±1.54 | 62.47±1.73 | 62.51±1.64 | 67.91±1.36 |
| | GraphPrompt | 63.12±1.24 | 62.56±1.31 | 61.94±1.06 | 68.36±1.04 |
| | All-in-one | 65.86±1.89 | 61.12±1.26 | 61.87±0.46 | 64.96±2.84 |
| | GPF | 67.16±1.21 | 62.84±1.69 | 62.17±1.04 | 68.11±1.66 |
| | GraphTOP | 68.94±2.06 | 65.12±2.34 | 65.27±1.49 | 69.16±1.31 |
| | EdgePrompt | 67.26±1.14 | 64.16±1.84 | 62.66±0.94 | 68.46±1.64 |
| | EdgePrompt+ | 68.16±1.56 | 67.96±1.44 | 68.26±0.91 | 70.16±0.66 |
| | **DiP-G (Ours)** | **70.36±1.24** | **69.81±1.11** | **70.46±0.86** | **72.61±0.74** |
| **SimGRACE** | Frozen+Linear | 64.16±1.04 | 63.66±1.76 | 63.56±2.09 | 67.26±1.34 |
| | GraphPrompt | 64.21±1.06 | 63.96±1.39 | 61.86±1.16 | 68.26±1.04 |
| | All-in-one | 69.16±0.66 | 60.26±2.19 | 63.16±0.56 | 64.86±2.11 |
| | GPF | 66.26±2.09 | 64.66±1.61 | 63.86±1.89 | 67.86±1.11 |
| | GraphTOP | 67.46±1.59 | 65.66±2.14 | 64.96±1.69 | 68.16±0.96 |
| | EdgePrompt | 66.36±2.34 | 65.46±1.51 | 63.96±1.74 | 68.56±0.84 |
| | EdgePrompt+ | 68.46±1.89 | 67.66±1.94 | 67.96±1.71 | 69.86±0.61 |
| | **DiP-G (Ours)** | **70.81±1.44** | **69.51±1.34** | **69.86±1.19** | **71.91±0.81** |
| **DP-GPPT** | Frozen+Linear | 62.96±2.04 | 58.86±1.16 | 63.56±0.76 | 66.86±1.34 |
| | GraphPrompt | 60.16±1.59 | 59.46±0.71 | 63.06±0.91 | 67.46±1.49 |
| | All-in-one | 62.84±1.31 | 59.36±1.54 | 62.46±1.06 | 65.36±0.91 |
| | GPF | 64.16±3.54 | 59.66±1.56 | 64.06±0.66 | 66.96±1.39 |
| | GraphTOP | **69.26±2.89** | 64.86±2.64 | 65.46±0.86 | 69.36±1.14 |
| | EdgePrompt | 65.16±3.34 | 60.96±1.64 | 63.96±0.71 | 67.56±1.46 |
| | EdgePrompt+ | 68.66±2.14 | 66.56±1.24 | 67.06±1.41 | 71.86±1.64 |
| | **DiP-G (Ours)** | 69.11±1.84 | **68.91±1.06** | **69.21±1.14** | **73.46±1.24** |
| **DP-GraphPrompt** | Frozen+Linear | 66.26±1.84 | 63.16±0.86 | 62.36±2.34 | 67.66±0.86 |
| | GraphPrompt | 66.56±1.74 | 63.26±1.04 | 62.66±1.11 | 67.96±0.74 |
| | All-in-one | 66.86±1.29 | 60.36±1.49 | 58.86±0.76 | 66.16±1.14 |
| | GPF | 68.06±2.74 | 63.06±1.51 | 63.36±1.74 | 68.36±1.04 |
| | GraphTOP | 68.56±4.04 | 64.26±1.06 | 63.96±2.49 | 68.16±1.06 |
| | EdgePrompt | 68.16±3.74 | 63.66±1.44 | 64.36±1.94 | 68.36±1.14 |
| | EdgePrompt+ | 69.46±3.04 | 67.16±0.61 | 66.26±1.21 | 71.16±1.94 |
| | **DiP-G (Ours)** | **71.21±2.09** | **69.46±0.74** | **68.81±1.04** | **73.06±1.39** |

hence $\|\Delta \mathbf{s}_b\|_2^2 \leq 2k$. Therefore,

$$\|\widehat{\mathbf{g}}_b\|_2^2 = \left\|\frac{\Delta \mathbf{s}_b}{\lambda_t}\right\|_2^2 \leq \frac{2k}{\lambda_t^2}. \tag{28}$$

Because the anchors use independent noise, $\{\widehat{\mathbf{g}}_b\}_{b=1}^B$ are independent conditional on $\boldsymbol{\Theta}$, and the variance of the sample mean scales as $1/B$. Applying this fact together with the uniform second-moment bound yields Eq. (27). □

## B. Overall Algorithm

The overall algorithm of DiP-G is provided in Algorithm 1.

## C. More experimental results

Table 5 reports results on four benchmark graph classification datasets under four pre-training strategies. DiP-G achieves the best performance in most settings and remains consistently competitive across all backbones. The gains are especially clear on the molecular datasets (NCI1, NCI109, and Mutagenicity), where learning discrete task-specific structural edits is important for capturing informative substructures. Compared with feature-prompting methods and prior topology/edge prompting baselines, DiP-G provides more reliable improvements across different pre-training objectives, suggesting good compatibility with both contrastive and link-prediction pre-training.

*Table 6.* Graph size and attribute information used in our experiments.

| Dataset | #Nodes | #Edges | #Features | Avg. Degree | Homophily | #Classes |
|---|---|---|---|---|---|---|
| Cora | 2,708 | 10,556 | 1,433 | 3.90 | 0.810 | 7 |
| PubMed | 19,717 | 88,648 | 500 | 4.49 | 0.802 | 3 |
| Amazon | 24,492 | 93,050 | 300 | 3.80 | 0.380 | 5 |
| Minesweeper | 10,000 | 39,402 | 7 | 3.94 | 0.683 | 2 |
| Flickr | 89,250 | 899,756 | 500 | 10.08 | 0.319 | 7 |

## D. More details about experimental setup

### D.1. Datasets

We adopt five real-world graph datasets to evaluate the performance of DiP-G. The statistics of these datasets can be found in Table 6.

### D.2. Pre-training Strategies

We evaluate DiP-G under four widely-used GNN pre-training pipelines that are also common in prior graph prompting studies. They cover both contrastive objectives and link-prediction objectives, allowing us to test the compatibility of our method across different sources of transferable representations.

**GraphCL** constructs two augmented views of the same input graph by applying stochastic graph transformations. A GNN encoder produces representations for both views, followed by a projection head that maps them to a contrastive space. Training minimizes a contrastive objective that increases the similarity of the two view representations, updating both the encoder and the projection head. **SimGRACE** forms a positive pair by coupling the representations from an original encoder and its perturbed counterpart. The perturbed encoder is obtained by injecting Gaussian noise into the model parameters. Given the same input graph, the two encoders generate two correlated representations, which are then optimized with a contrastive loss. **DP-GPPT** randomly removes a portion of edges and trains the model to recover the missing connectivity. The objective predicts whether a node pair should be connected, and negative pairs are sampled from node pairs that are not connected in the original graph. **DP-GraphPrompt** is another link-prediction style objective based on per-node sampling. For each anchor node, we sample a connected neighbor as a positive example and an unconnected node as a negative example. The model is trained to assign higher similarity to positive node pairs than to negative ones.

Developing stronger pre-training objectives for GNNs is an active research topic. Existing graph prompting works often rely on either contrastive learning or link prediction, and recent studies also discuss when each objective is preferable. In this paper, we do not aim to propose a new pre-training strategy. Instead, our focus is on effective downstream adaptation, given a frozen pre-trained GNN encoder , we learn task-specific prompts to improve few-shot performance.

### D.3. Baselines

We compare DiP-G with a set of strong prompting baselines that cover both feature-side and structure-side adaptation. All methods use the same frozen backbone encoder and follow the same few-shot protocol for a fair comparison.

**Frozen+Linear.** We keep the pre-trained encoder fixed and train a linear classifier on top of node representations. This serves as a simple reference without any prompting. **GPPT** adapts a frozen encoder by reformulating node classification into a link-prediction style objective, aiming to better match common pre-training signals. **All-in-One** provides a unified prompting framework that supports multiple downstream tasks and learns lightweight prompts while keeping the backbone frozen. **GraphPrompt** learns feature-level prompts to adapt pre-trained GNNs. **GraphPrompt+** strengthens the original design with an enhanced prompting module and training recipe. **ProNoG** focuses on adaptation in challenging graphs by designing prompts that better align with heterophily patterns. **GraphTOP** performs topology-oriented prompting by directly modifying the graph structure to adapt a frozen GNN encoder. **EdgePrompt** learns edge-level prompts that modulate message passing on selected edges. **EdgePrompt+** is a stronger variant with improved edge prompting and optimization settings.

For each baseline, we follow the official implementations or the settings reported in the original papers whenever applicable, and only tune hyperparameters when explicitly stated.

