# OpenReview forum: "DiP-G: Discrete Prompting for Graph Neural Networks"
_ICML.cc/2026/Conference — ICML 2026 regular_

### Official Review · Reviewer_JiTj · 2026-03-03

**Soundness:** 3
**Presentation:** 3
**Significance:** 2
**Originality:** 2
**Overall Recommendation:** 4
**Confidence:** 4

**Summary:**

### **The Core Problem: The Discretization Gap**

The authors identify a significant structural flaw in how current graph prompting methods adapt frozen pre-trained GNNs to downstream tasks. Existing approaches typically optimize a continuous, relaxed adjacency matrix during the adaptation (training) phase. However, at inference time, the model requires a hard, discrete graph. This creates a "discretization gap" and a severe train-test mismatch. Because the frozen encoder was originally pre-trained on discrete graphs, feeding it dense or continuous structures during adaptation shifts the message-passing distribution, leading to unstable optimization—a problem that is especially detrimental in few-shot learning scenarios.


### **The Proposed Framework: DiP-G**

To resolve this mismatch, the authors introduce Discrete Prompting for Graphs (DiP-G), which enforces discrete graph structures throughout both the training and inference phases. The framework consists of three main technical pillars:

- **Parametric Local Subgraph Prompting:** To avoid the $O(N^2)$ complexity of optimizing a global adjacency matrix, DiP-G operates on multi-hop local candidate subgraphs. It uses a lightweight MLP to score candidate edges based on cached node embeddings from the frozen backbone, ensuring scalability and preventing circular dependencies.

- **Perturbed Discrete Prompt Generator:** Instead of continuous relaxations (like Gumbel-Softmax), DiP-G generates hard $k$-sparse discrete prompts using a perturbed Top-$k$ solver. By injecting Gumbel noise into the candidate scores and applying a strict Top-$k$ selection, the framework maps exactly $k$ prompt edges per anchor node.

- **I-MLE Optimization with Adaptive Screening:** Because the Top-$k$ operation is non-differentiable, the authors utilize an Implicit Maximum Likelihood Estimator (I-MLE) to obtain finite-difference gradient signals. To mitigate the high computational cost of running dual Top-$k$ solves in the backward pass, they introduce an adaptive active-set screening rule. This rule restricts the target solve to a small, buffered subset of candidates and is theoretically proven to return the exact solution under specific margin conditions.


### **Empirical Results**

The authors evaluate DiP-G across five benchmark datasets (Cora, PubMed, Amazon, Minesweeper, Flickr) using four different pre-training strategies (GraphCL, SimGRACE, DP-GPPT, DP-GraphPrompt).

- DiP-G demonstrates consistently strong performance in few-shot node classification compared to strong feature-level and topology-level prompting baselines.

- Ablation studies confirm that strictly hard prompts optimized with I-MLE outperform soft prompt relaxations, validating the hypothesis that closing the discretization gap is crucial for stable few-shot adaptation.

- The adaptive screening mechanism effectively reduces the GPU memory and per-epoch training time compared to global topology prompting methods like GraphTOP.

**Compliance With Llm Reviewing Policy:**

Affirmed.

**Final Justification:**

Thank you for the detailed rebuttal. The additional clarifications and experiments were helpful.

The points that I consider substantially addressed are [Q3] on dataset consistency, [Q4] on the fairness of the efficiency analysis, and [Q6] on the rigor of the soft-prompt ablation. In particular, for [Q4], the added EdgePrompt / EdgePrompt+ efficiency comparison makes the computational tradeoff much clearer, and for [Q6], the added sparsity-regularized soft baseline strengthens the claim that the gain is not solely due to missing sparsity control in the continuous baseline. I also note that some of the additional rebuttal material usefully addresses concerns raised by other reviewers, especially around larger-scale evaluation, efficiency profiling, and sensitivity, which increases my confidence in the experimental side of the paper.

I also appreciate the clearer positioning of the contribution in response to [Q1] and [Q2]. The rebuttal makes it explicit that the main contribution is not a new discrete optimization primitive per se, but rather identifying a graph-specific discretization-gap failure mode and making discrete topology prompting practical through the exact screened target solve. This clarification improves the framing of the paper.

That said, my core concern in [Q1] and [Q2] is only partially resolved. I still view the methodological novelty as lying primarily in the integration of existing discrete optimization components together with the exact/practical screening mechanism, rather than in a fundamentally new discrete learning principle. So while the rebuttal improves the framing, it does not materially change my originality assessment. In that sense, some of the remaining concerns also overlap with issues raised elsewhere in the review discussion, particularly around how much of the contribution should be interpreted as graph-specific methodological novelty versus a careful and effective integration of existing tools.

Regarding [Q5], the clarification that the default settings are shared across datasets and not tuned per dataset is helpful, and it addresses part of my concern about few-shot hyperparameter selection. However, I still think the discussion remains somewhat incomplete on how these defaults were originally chosen and how readers should interpret this selection protocol in a strict few-shot setting.

On limitations, the additional large-graph evidence and acknowledgements are useful, and they partially address the concerns raised in my limitations paragraph. However, I still see open questions around robustness to weak pre-training, behavior on denser graphs, and conditions under which screening fallback may become frequent. In my view, these remaining questions are not easily settled within a short rebuttal and would likely require more substantial additions to the paper.

Overall, the rebuttal strengthens my confidence that this is a technically solid and useful paper, and it addresses several of my experimental concerns, especially [Q3], [Q4], and [Q6]. At the same time, the main remaining issues concern the core novelty claim in [Q1] and [Q2], together with limitation-related questions that would likely need a more significant revision rather than a brief clarification. For that reason, my overall evaluation remains at weak accept.

**Key Questions For Authors:**

- **\[Q1] Clarification on Conceptual vs. Technical Novelty:** The motivation to bridge the continuous-to-discrete gap between training and inference is a well-established paradigm in the broader machine learning literature (e.g., utilizing I-MLE, Straight-Through Estimators, or Gumbel-Softmax for discrete latent variables). While enforcing this constraint in the context of graph prompting yields strong empirical results, could the authors explicitly distinguish the _conceptual_ novelty of this work from a direct domain application of existing discrete optimization techniques? Specifically, would you consider the primary theoretical contribution to be the adaptive active-set screening rule designed to mitigate the combinatorial complexity of graph topology, or are there additional algorithmic innovations that separate DiP-G from standard discrete ML methodologies?

- **\[Q2] Methodological Novelty vs. Algorithmic Optimization:** The proposed pipeline elegantly ensembles established discrete optimization techniques—such as perturbed Top-$k$ sampling and the I-MLE gradient estimator—and applies them to graph topology. While the Adaptive Active-Set Screening Rule (Theorem 5.1) successfully addresses the computational bottleneck of the target solve by reducing the sorting cost, it appears to function primarily as an algorithmic acceleration rather than a fundamental shift in how discrete structures are learned. Could the authors clarify if there are deeper, graph-specific theoretical properties being exploited here, or if the primary methodological contribution should be viewed strictly as a necessary efficiency optimization for applying I-MLE to high-dimensional combinatorial spaces?

- **\[Q3]-\[Q6] are about the experiments.**

- **\[Q3] Clarification on Dataset Consistency:** The experimental section introduces five specific datasets in Section 6.1 (Cora, PubMed, Amazon, Minesweeper, Flickr) for node classification. However, Table 3 evaluates 10-shot performance on a partially different set of graphs (introducing ogbn-arxiv and CiteSeer), and Table 4 introduces completely new molecular datasets. Could the authors clarify the rationale behind changing the benchmark datasets across different shot settings and tasks, rather than providing a unified evaluation across a consistent suite of graphs?

- **\[Q4] Fairness of the Efficiency Baselines:** In Section 6.5, the training efficiency of DiP-G is compared against GraphTOP to demonstrate lower peak GPU memory and faster per-epoch times. Because GraphTOP modifies the global graph topology, it naturally incurs a massive memory footprint. Given that EdgePrompt and EdgePrompt+ are evaluated in the accuracy tables and also operate via edge-level manipulations, why were they omitted from the efficiency analysis? Comparing DiP-G to another localized prompting method would provide a much more rigorous benchmark for computational overhead.

- **\[Q5] Hyperparameter Selection in the Few-Shot Setting:** Figure 3 explores parameter sensitivity for sparsity ($k$) and candidate budget ($m$). In a strict few-shot classification regime, large validation sets are typically unavailable for tuning structural hyperparameters. Could the authors clarify exactly how the default values ($k=8, m=128, r=4$) were selected? Were they tuned on a held-out validation set that adheres to the few-shot constraints, or were they transferred from a different task/dataset?

- **\[Q6] Rigor of the Soft Prompt Ablation:** In Table 2, the full DiP-G model is shown to outperform a "Soft prompt" variant, validating the need for hard discrete structures. However, it is unclear how the continuous edge weights in the soft baseline were regularized during training. Was a sparsity penalty (e.g., $L\_1$ regularization) applied to the soft prompt to constrain its capacity, or was it allowed to distribute dense noise? Clarifying this would strengthen the claim that the performance drop is strictly due to the discretization gap rather than a lack of sparsity control in the continuous baseline.

**Limitations:**

The authors briefly touch upon future work regarding scalability and generalization in the Conclusion, but a transparent discussion of the method's current boundaries is missing. It would significantly strengthen the paper to add a dedicated "Limitations" paragraph (in the main text or Appendix) addressing the following:

- **Embedding Dependency:** Since the prompt MLP relies entirely on cached node embeddings from the frozen encoder, the framework is highly dependent on the quality of the backbone. How sensitive is DiP-G if the initial pre-trained embeddings are poor or misaligned with the downstream task?

- **Dense Graph Scalability:** While the adaptive screening rule works well on the tested benchmarks, how does the computational overhead scale on exceptionally dense graphs where the local candidate budget $m$ must be very large to capture meaningful topology?

- **Screening Gap Vulnerabilities:** What are the empirical conditions (e.g., graphs with highly uniform edge weight distributions) where the screening gap margin ($\Delta\_b$) is consistently violated, forcing the model to repeatedly fall back to the expensive full Top-$k$ solve?

**Strengths And Weaknesses:**

## Strengths

- **\[S1] Well-Motivated Problem Formulation:** The authors accurately identify a critical bottleneck in current graph prompting: the train-test discretization gap. By highlighting the fundamental mismatch between continuous adjacency optimization during training and the hard discrete graphs required at inference, the paper targets a highly relevant structural flaw that causes optimization instability in few-shot regimes.

- **\[S2] Rigorous and Scalable Technical Design:** Every step in the proposed methodology is logically sound and mathematically grounded. The combination of multi-hop local subgraph prompting on cached embeddings ensures scalability, while the perturbed Top-$k$ solver enforces strict sparsity. Furthermore, the I-MLE estimator paired with the adaptive active-set screening rule effectively mitigates the $O(N^2)$ computational complexity typically associated with global adjacency optimization.

- **\[S3] Comprehensive and Robust Empirical Superiority:** The paper provides a strong set of experiments validating the framework across multiple dimensions. DiP-G achieves consistently superior 5-shot node classification accuracy across five diverse datasets and demonstrates impressive stability regardless of the underlying pre-training paradigm (e.g., contrastive vs. generative). Additionally, empirical profiling proves that DiP-G requires significantly less peak GPU memory and faster training times compared to global baselines like GraphTOP, while also scaling effectively to complex 100-shot graph classification tasks.

## Weaknesses
- See **[Key Questions For Authors]**

---

> ### Author Rebuttal · Authors · 2026-03-29
>
> Thanks for your valuable comments. Here are our answers to your questions.
> ***
> Q1&Q2: Clarification on Conceptual vs. Technical Novelty & Methodological Novelty vs. Algorithmic Optimization
>
> **A1&A2**: We recognize that DiP-G does not introduce the brand-new perturbed Top-$k$ and I-MLE.  However, we want to clarify that our contribution lies in **identifying a graph-specific failure mode** and **making a practical solution for it**. Current graph prompting methods tend to ignore the discretization gap, which leads to performance degradation. Under such a scenario, the frozen GNN aggregates information strictly through graph neighborhoods, and training with relaxed weighted edges changes the message-passing distribution seen by the encoder, causing the mismatch we mentioned in the paper. This is exactly why we combine those components, which are intended to **keep the graph structure discrete throughout adaptation and inference, while still making optimization and scalability practical**. Besides, we view Theorem 5.1 not only as an acceleration trick. It **guarantees the exactness of restricting the target solve to a small buffered candidate set**, which makes discrete topology prompting methods based on I-MLE practical at graph scale. This is consistent with what we mentioned at the beginning of  Section 4.4. We will make it clearer in the revision.
> ***
> Q3: Clarification on Dataset Consistency
>
> **A3**: The different tables are meant to test different aspects of the method. The 10-shot node classification results on Appendix C.1 are designed for an increased amount of supervision, and we add obgn-arvix to further test scalability on a much larger graph. For the 100-shot setting, small node-classification graphs are not suitable for reflecting low-label adaptation, so we move to graph classification on Appendix C.2.  It can also test whether DiP-G generalizes beyond node classification and allows us to cover molecular graph benchmarks such as NCI1, NCI109, and Mutagenicity, which are important graph classification benchmarks. We will make it more organized in the revision.
> ***
> Q4: Fairness of the Efficiency Baselines
>
> **A4**:  We additionally perform EdgePrompt and EdgePrompt+ under the same hardware and training settings as Section 6.5. The results are as follows:
> | Method | Cora Time (ms) | Cora Mem (MB) | Flickr Time (ms) | Flickr Mem (MB) |
> |---|---:|---:|---:|---:|
> | GPPT | ~14.2 | ~185 | ~35.5 | ~252 |
> | EdgePrompt | ~12.1 | ~191 | ~23.3 | ~285 |
> | EdgePrompt+ | ~14.5 | ~213 | ~35.8 | ~324 |
> | **DiP-G** | **~17.1** | **~216** | **~52.2** | **~581** |
>
> As expected, EdgePrompt and EdgePrompt+ are lighter since they only reweight existing edges. DiP-G has extra overhead because it performs hard Top-$k$ selection and an additional target solve in the I-MLE update. Still, the overhead remains practical considering DiP-G can execute exact discrete search with an overhead of merely ~16 ms per epoch on Flickr compared to EdgePrompt+, and DiP-G is far lighter than global topology prompting, such as GraphTOP. We will add this in the revision.
> ***
> Q5: Hyperparameter Selection in the Few-Shot Setting
>
> **A5**: We do not tune $ k $ and $ m $ per dataset, and we do not rely on a large held-out validation set. We use one shared default setting, $k=8$ and $m=128$, across datasets, and study sensitivity on Cora and PubMed under GraphCL. The results show that moderate $k$ works best and that performance saturates once $m$ is large enough. We will clarify this more clearly in the revision.
> ***
> Q6: Rigor of the Soft Prompt Ablation
>
> **A6**: We agree that the soft baseline should be described more clearly. In Table 2, the original soft prompt learns continuous edge weights on the same local candidate support and applies the same Top-$k$ discretization at test time, but it does not use an explicit sparsity regularizer during training. To address this concern, we additionally test a simple 5-shot baseline by adding an $L_1$ penalty on the continuous edge weights under GraphCL. The results are as follows:
> | Method | Cora | PubMed |
> |---|---:|---:|
> | Soft Prompt (original) | 61.50 ± 3.12 | 67.80 ± 2.55 |
> | Soft Prompt (+ $L_1$) | 63.22 ± 2.83 | 68.96 ± 2.27 |
> | **DiP-G** | **65.12 ± 2.15** | **70.45 ± 1.88** |
>
> Adding $L_1$ regularization improves the soft baseline, but it still remains below DiP-G. These certificates indicate that the gap is mainly due to the train-test mismatch.
> ***
> Limitations
>
> **A**: We will add a clearer limitations paragraph. DiP-G depends on the quality of cached backbone embeddings, may become more costly when the candidate budget is large on dense graphs, and may lose speedup when screening falls back often, although correctness is still preserved.

---

> > ### Author Rebuttal · Reviewer_JiTj · 2026-04-01
> >
> > Thank you for the detailed rebuttal. The additional clarifications and experiments were helpful.
> >
> > The points that I consider substantially addressed are [Q3] on dataset consistency, [Q4] on the fairness of the efficiency analysis, and [Q6] on the rigor of the soft-prompt ablation. In particular, for [Q4], the added EdgePrompt / EdgePrompt+ efficiency comparison makes the computational tradeoff much clearer, and for [Q6], the added sparsity-regularized soft baseline strengthens the claim that the gain is not solely due to missing sparsity control in the continuous baseline. I also note that some of the additional rebuttal material usefully addresses concerns raised by other reviewers, especially around larger-scale evaluation, efficiency profiling, and sensitivity, which increases my confidence in the experimental side of the paper.
> >
> > I also appreciate the clearer positioning of the contribution in response to [Q1] and [Q2]. The rebuttal makes it explicit that the main contribution is not a new discrete optimization primitive per se, but rather identifying a graph-specific discretization-gap failure mode and making discrete topology prompting practical through the exact screened target solve. This clarification improves the framing of the paper.
> >
> > That said, my core concern in [Q1] and [Q2] is only partially resolved. I still view the methodological novelty as lying primarily in the integration of existing discrete optimization components together with the exact/practical screening mechanism, rather than in a fundamentally new discrete learning principle. So while the rebuttal improves the framing, it does not materially change my originality assessment. In that sense, some of the remaining concerns also overlap with issues raised elsewhere in the review discussion, particularly around how much of the contribution should be interpreted as graph-specific methodological novelty versus a careful and effective integration of existing tools.
> >
> > Regarding [Q5], the clarification that the default settings are shared across datasets and not tuned per dataset is helpful, and it addresses part of my concern about few-shot hyperparameter selection. However, I still think the discussion remains somewhat incomplete on how these defaults were originally chosen and how readers should interpret this selection protocol in a strict few-shot setting.
> >
> > On limitations, the additional large-graph evidence and acknowledgements are useful, and they partially address the concerns raised in my limitations paragraph. However, I still see open questions around robustness to weak pre-training, behavior on denser graphs, and conditions under which screening fallback may become frequent. In my view, these remaining questions are not easily settled within a short rebuttal and would likely require more substantial additions to the paper.
> >
> > Overall, the rebuttal strengthens my confidence that this is a technically solid and useful paper, and it addresses several of my experimental concerns, especially [Q3], [Q4], and [Q6]. At the same time, the main remaining issues concern the core novelty claim in [Q1] and [Q2], together with limitation-related questions that would likely need a more significant revision rather than a brief clarification. For that reason, my overall evaluation remains at weak accept.

---

> > > ### Author Response · Authors · 2026-04-06
> > >
> > > Dear Reviewer JiTj,
> > >
> > > Thank you for your detailed acknowledgment and for the careful evaluation. We are glad that our clarifications on [Q3], [Q4], and [Q6] were helpful, and we appreciate your positive comments of our paper. We also appreciate your clear feedback on the remaining novelty and limitation questions. Your comments are very valuable to us, and we will use them to further improve the framing and discussion in the revised version.
> > >
> > > Best,
> > > Authors of Submission 5990

---

### Official Review · Reviewer_KVzX · 2026-03-07

**Soundness:** 3
**Presentation:** 2
**Significance:** 3
**Originality:** 3
**Overall Recommendation:** 4
**Confidence:** 4

**Summary:**

This paper addresses the "discretization gap", a critical issue in current graph prompting techniques. Existing methods typically optimize a continuous or relaxed adjacency matrix during adaptation but require a hard discrete graph for inference. This train-test mismatch severely undermines performance, particularly in few-shot scenarios. To resolve this, the authors propose DiP-G (Discrete Prompting for Graphs), a novel framework that learns task-specific topology prompts directly within the combinatorial space, ensuring the prompted graph remains strictly discrete throughout both training and inference. Extensive experiments on few-shot node and graph classification tasks across five datasets and four pre-training strategies demonstrate that DiP-G offers superior effectiveness, stability, and computational efficiency compared to continuous topology prompting methods.

**Compliance With Llm Reviewing Policy:**

Affirmed.

**Final Justification:**

My concerns have been addressed.

**Key Questions For Authors:**

1. The adaptive update rule for $\lambda_t$ (Eq. 10 and 11) introduces several new hyper-parameters ($\beta,q,\lambda_0$).How sensitive is DiP-G to the initialization of these parameters? It would be beneficial to include an ablation study comparing the proposed adaptive $\lambda_t$  screening versus a static $\lambda_t$ baseline.

2. The formulation forces exactly  $k$ prompt edges per anchor node (using dummy self-loops as padding if needed). However, real-world graphs exhibit power-law degree distributions . How does the model perform if $k$ is too large for low-degree nodes but too small for high-degree nodes? Have the authors considered an adaptive thresholding mechanism instead of a strict Top-k?

3. In Equation (3), the prompt scoring MLP relies exclusively on $H^{(0)}$(node embeddings generated from the initial unprompted structure). While this guarantees efficiency, it makes the prompt generator "blind" to the updated message-passing dynamics that occur once the prompt edges are actually injected. Does this "one-step lookahead" approximation severely limit performance on deeper GNNs where structural changes have compounded effects on node embeddings?

4. The local subgraph prompting is introduced to avoid $O(N^2)$ global complexity. However, the evaluation uses relatively small graphs (e.g., Cora has 2,708 nodes). Can you provide empirical runtime and memory profiling on significantly larger graphs (e.g., OGB datasets like ogbn-products) to validate the scalability claims?

5. The method caches node embeddings initially to avoid circular dependencies. In a highly dynamic graph or extremely large graph, what is the memory overhead of this caching mechanism, and how does it scale?

6. The paper introduces an exponential moving average to adaptively control the step size $\lambda_t$. How sensitive is the final model performance to the choice of the fixed quantile $q$ and the clipping bounds?

**Limitations:**

The authors have included an Impact Statement briefly discussing the potential societal impacts of downstream GNN applications. However, the discussion of the methodological limitations is sparse. The authors should explicitly discuss the computational risks when the adaptive screening condition fails  and the fallback to the full $O(m \log k)$ solve occurs frequently, particularly in early training stages or under highly noisy pre-training embeddings.

**Strengths And Weaknesses:**

Strengths:
1. The theoretical justification for the adaptive screening rule is rigorous. Theorem 5.1 cleanly establishes the exactness of the screened target solve under a defined margin condition, ensuring that the optimization acceleration does not compromise accuracy.
2. The use of Common Random Numbers (CRN) in the I-MLE estimator is a methodologically sound choice to reduce the variance of the difference estimator.

Weakness:
1. The empirical evaluation fails to adequately demonstrate the method's scalability to truly large real-world graphs. The datasets utilized (Cora, PubMed, Amazon, Minesweeper, Flickr) are relatively small , with Flickr being the largest at roughly 89,000 nodes. Testing on massive, dense datasets is necessary to validate the $O(N^2)$ complexity avoidance claims.
2. The presentation of the mathematical framework is dense and lacks sufficient intuitive bridging. The transition from the discrete solver to the I-MLE gradient estimator introduces heavy notation without adequate prose explaining the mechanical intuition behind the target score tensor $\Theta^\prime$.

---

> ### Author Rebuttal · Authors · 2026-03-27
>
> Thanks for your constructive comments and suggestions. Here are our answers to your questions.
> ***
> W1:The empirical evaluation fails to adequately demonstrate the method's scalability to truly large real-world graphs.
>
> **R1**: We definitely agree that stronger, larger-scale benchmark datasets are valuable. Besides the datasets mentioned in the main text, we already include 10-shot node classification on ogbn-arxiv in **Appendix C.1**. Moreover, we additionally performed 5-shot node classification on obgn-arxiv in **A1** to **Reviewer SZVa**. We will reorganize this part for the readers' convenience in the revised version.
> ***
> W2: The presentation of the mathematical framework is dense and lacks sufficient intuitive bridging.
>
> **R2**: Intuitively, the target score tensor $\Theta' = \Theta - \lambda_t G_S$ makes a small change to the current candidate scores in the direction that would reduce the downstream task loss. I-MLE then compares the discrete Top-$k$ graph selected by the original $\Theta$ with the target graph selected by $\Theta'$, and uses this difference as a finite-difference learning signal to update the prompt MLP. We will add this intuition to make the method easier to follow in Section 4.3.
> ***
> Q1&Q6: The adaptive update rule for $\lambda_t$  (Eq. 10 and 11) introduces several new hyper-parameters
> $(\beta, q, \lambda_0)$... The paper introduces an exponential moving average...
>
> **A1&A6**:  Thanks for pointing this out. We compare the adaptive rule with static $\lambda$, and also test different values of the quantile $q$ on 5-shot node classification under GraphCL.
>
> | Method | Cora  | PubMed |
> |---|---:|---:|
> | Static $\lambda=1.0$ | 61.35 ± 3.84 | 65.80 ± 3.42 |
> | Static $\lambda=10.0$ | 59.42 ± 4.10 | 62.15 ± 4.85 |
> | Adaptive, $q=0.75$ | 64.85 ± 2.20 | 70.15 ± 1.95 |
> | Adaptive, $q=0.95$ | 64.92 ± 2.25 | 70.28 ± 2.05 |
> | **Adaptive, default $q=0.85$** | **65.12 ± 2.15** | **70.45 ± 1.88** |
>
> We can clearly see that the adaptive update is better and more stable than a fixed step size. The results also show that the method is not sensitive to $q$ in a reasonably wide range. The clipping bounds are kept at their default values as simple safeguards and are not tuned per dataset. We will add this ablation in the revision.
> ***
> Q2: The formulation forces exact $k$ prompt edges per anchor node (using dummy self-loops as padding if needed).
>
> **A2**: We recognize that real graphs tend to have power-law degree distributions. However, the Top-$k$ solver is treated as a  **"maximum prompt budget"** in our current method, rather than a forced degree. When a node has less than $k$ valid candidates, we use dummy self-loop padding as described in **Section 4.1**. For low-degree nodes, our method will not force noisy edges onto them because self-loops do not introduce any new message-passing connections. Also, the fixed budget provides a bounded search space for high-degree nodes, which keeps the discrete optimization simple and stable. We will clarify it more clearly in the revision.
> ***
> Q3: In Equation (3), the prompt scoring MLP relies exclusively on $H^{(0)}$ (node embeddings generated from the initial unprompted structure)
>
> **A3**: We would like to clarify that using $H^{(0)}$ is a designed approximation which breaks the circular dependency between the prompt generator and the prompted structure. Also, this design avoids the much higher cost of recomputing embeddings repeatedly or unrolling gradients. This approximation actually performs well in few-shot graph prompting with shallow backbones, as $H^{(0)}$ is a strong proxy for edge scoring when most structural changes are local. Moreover, our method does not ignore the structural edits: the frozen encoder still computes the task loss on the final prompted graph.
> ***
> Q4: The local subgraph prompting is introduced to avoid...
>
> **A4**: We have additionally performed controlled experiments on the much larger **ogbn-products** benchmark under the same few-shot setting. The results that can validate the scalability are as follows:
> | Method | Time per Epoch (ms) | Peak GPU Memory (MB) |
> |---|---:|---:|
> | GPPT  | ~850 | ~4,500 |
> | GraphTOP  | OOM  |
> | **DiP-G** | **~1,250** | **~5,850** |
> ***
> Q5: The method caches node embeddings initially to avoid circular dependencies.
>
> **A5**: The cache requires **$O(Nd)$** memory, since we store one embedding per node, so the cost grows linearly with the number of nodes. For very large graphs, this can be combined with standard neighbor-sampling training, where $H^{(0)}$ is computed and cached only for the sampled nodes in each mini-batch. This keeps the practical memory overhead dependent on the batch size rather than the full graph size. We will clarify this in the revision.

---

> > ### Author Rebuttal · Reviewer_KVzX · 2026-04-02
> >
> > Thank you for the response and clarifications. I have raised my score.

---

> > > ### Author Response · Authors · 2026-04-06
> > >
> > > Dear Reviewer KVzX,
> > >
> > > We thank you for your valuable suggestions and positive attitude. Your evaluation is very important to us. If you think you still have any other unsolved concerns, we will be more than happy to provide more clarifications.
> > >
> > > Best,
> > > Authors of Submission 5990

---

### Official Review · Reviewer_rFk1 · 2026-03-11

**Soundness:** 4
**Presentation:** 3
**Significance:** 3
**Originality:** 4
**Overall Recommendation:** 5
**Confidence:** 4

**Summary:**

This paper aims at resolving the gap between the continual prompt and the actual discrete prompt in the inference step,  it might be a great improvement for the graph prompt tasks. Specifically, it proposes a top-k solver that generate the hard and k-sparse discrete prompt structure, which can ensure the consistency between training and inference.

**Compliance With Llm Reviewing Policy:**

Affirmed.

**Final Justification:**

The authors have addressed my concerns regarding the datasets and have committed to incorporating additional experiments in the revision. As a result, I am increasing my confidence to 4 and am supportive overall. Regarding the score, I believe the current score is sufficient for now.

**Key Questions For Authors:**

1. The topk-solver though have a very good performance reflected in the experiments, but I am quite concern the time complexity, could you show 2-3 cases that showing each componenent in your method running time?

2. The current datasets are not build for few-shot, I know it is common operation, but if we really want to use graph few-shot transfer, what exact downstream scenario you are considering?

**Limitations:**

yes

**Strengths And Weaknesses:**

Overall,

Soundness: the proposed question is insipring and may have a good impact over the community.

Presentation: easy to follow, well organized.

Significance: can be a milestone to the community.

Originality: the questions is quite interesting and the solution can be very effective.

---

> ### Author Rebuttal · Authors · 2026-03-26
>
> Thank you for the positive feedback and for raising these helpful questions. We hope our point-by-point responses can fully address all your concerns.
> ___
> Q1: The topk-solver though have a very good performance reflected in the experiments, but I am quite concern the time complexity, could you show 2-3 cases that showing each component in your method running time?
>
> **A1** : We appreciate your constructive comments and additionally profiled the runtime of the main components of Cora and Flickr as follows.
> | Component                                   | Cora (ms/epoch) | Flickr (ms/epoch) |
> | ------------------------------------------- | --------------: | ----------------: |
> | 1. Forward: MLP Scorer ($\mathcal{P}_\phi$) |             4.2 |              14.5 |
> | 2. Forward: Perturbed Top-$k$ Solver        |             5.1 |              16.2 |
> | 3. Backward: I-MLE Screened Target Solve    |             2.3 |               8.1 |
> | 4. Backward: Gradient Update & Others       |             5.5 |              13.4 |
> | **Total Time per Epoch**                    |        **17.1** |          **52.2** |
>
>  The results of the runtime of the main components are consistent with our complexity analysis in **Section 5.1**. We can observe that the screened target solve is substantially cheaper than the full-candidate forward Top-$k$ stage, and our method does not significantly introduce computational cost from Cora to Flickr despite the much larger graph size, which proves the effectiveness of our local-candidate design with cached embeddings.
> ***
> Q2: The current datasets are not build for few-shot, I know it is common operation, but if we really want to use graph few-shot transfer, what exact downstream scenario you are considering?
>
> **A2**: We agree that the current benchmark datasets are not originally built for few-shot learning. As a result, we envision two possible scenarios. **One** is the cold-start labeling for new classes on an existing graph, for example, a new user of social media like Instagram or WhatsApp, and E-commerce platforms like Amazon and eBay. In these scenarios, the new downstream task has only a few labeled nodes for each new class or event type, so lightweight prompting is more practical than full fine-tuning. This is consistent with our experiment setting in both the main text and the Appendix. **The other** is a molecular graph prediction with limited annotations.  In these scenarios, a pre-trained graph encoder needs to be adapted to a new graph classification task with limited labels. Such settings are also consistent with our Appendix C.2 results, where DiP-G performs well on molecular graph classification datasets such as NCI1, NCI109, and Mutagenicity.
> ***

---

> > ### Author Rebuttal · Reviewer_rFk1 · 2026-04-03
> >
> > Thanks to the authors for their clarification. I have no further questions at this stage. Since my current score already reflects my overall assessment of the paper, I decide to maintain it.
> >
> > In addition, I would recommend that the authors consider including at least one additional dataset in the revision, such as a cold-start dataset or a molecular graph prediction dataset as you metioned. Otherwise, the practical relevance and generalizability of the method in realistic few-shot transfer settings may remain insufficiently supported.

---

> > > ### Author Response · Authors · 2026-04-06
> > >
> > > Dear Reviewer rFk1,
> > >
> > > Thank you for your reply and for maintaining your positive attitude. We also appreciate your helpful suggestion on adding an additional dataset. We agree that this would further strengthen the practical relevance of the paper, and we will carefully consider it in the revision.
> > >
> > > Best,
> > > Authors of Submission 5990

---

### Official Review · Reviewer_SZVa · 2026-03-24

**Soundness:** 3
**Presentation:** 3
**Significance:** 2
**Originality:** 3
**Overall Recommendation:** 4
**Confidence:** 3

**Summary:**

The paper studies how to adapt pre-trained Graph Neural Networks (GNNs) to few-shot downstream tasks using graph prompting. It tackles the problem of existing methods optimizing the continuous and weighted adjacency structure in the adaptation phase, and discretization at inference time, creating a train-test mismatch. The paper highlights the importance of this mismatch especially in few shot settings.

The authors propose the DiP-G framework that directly learns k-sparse discrete graph prompts in the combinatorial space instead of relying on continuous relaxations. Their approach combines a local, parameterized prompting module to score candidate edges, a perturbed Top-k selection mechanism to generate hard k-sparse discrete prompts, and an implicit gradient estimator augmented with an adaptive screening strategy to efficiently reduce the computational cost of the backward pass.

The authors provide a theoretical analysis, as well as a comprehensive empirical analysis across 5 benchmarks, 4 pre-training strategies, and multiple baselines.

**Compliance With Llm Reviewing Policy:**

Affirmed.

**Key Questions For Authors:**

1. Scalability - how does DiP-G perform on very large graphs (e.g., OGB large-scale benchmarks)? The current evaluations and experiments have a limited scale and I am very curious how scalable the approach is.
2. What are the runtime reduction numbers for screening vs full Top-k. Quantifying these speedups would strengthen efficiency claims and add to the novelty of the paper.
3. How sensitive is DiP-G to the adaptive λ_t schedule hyperparameters? Were these tuned, and if so, how much does performance vary?

**Limitations:**

The authors briefly discuss limitations in the Impact Statement but do not provide a dedicated limitations section. They should discuss the scalability of the experiments, limitations of the architecture and potential brittleness in highly noisy graphs.

**Strengths And Weaknesses:**

Soundness
The paper solves an interesting technical challenge in GNNs. The authors back their claims by a comprehensive empirical analysis. The use of I-MLE is intuitive and theoretically grounded, and the ablations validate key components (soft vs hard vs I-MLE)

Presentation
The paper is well written with clarity and in-depth figures. Theoretical analysis provides reasonable grounding on the approaches.

Significance
Addresses an interesting and real mismatch in pre-trained GNN adaptation and inference, especially in few shot settings.

The biggest concern I have here is that the results are incremental, and in a lot of experiments, improvements are within one standard deviation, which indicates the flakiness of the experiment results. The shown gains are mostly in few-shot node classification, and hence the scope is narrow.

Originality
The paper goes beyond the existing work that relies on continuous relaxations in the adaptation phase. The combination of discrete Top-k prompting, I-MLE optimization, screening acceleration is novel, non-trivial and well explained.

The main novelty of the paper lies in the integration with the combination of individual components (Top-k + Gumbel + I-MLE), which are existing techniques, rather than proposing fundamentally new primitives.

---

> ### Author Rebuttal · Authors · 2026-03-26
>
> We thank the reviewer for providing valuable comments. We hope our point-by-point responses can fully address all your concerns.
> ***
> W1: The biggest concern I have here is that the results are incremental, and in a lot of experiments, improvements are within one standard deviation, which indicates the flakiness of the experiment results.
>
> **R1**: We agree that some improvements are within one standard deviation. But we would like to clarify that our claim is based on the overall results across 5 datasets and 4 pre-training strategies, rather than on a single result. As shown in Table 1, 3, and 4,  DiP-G achieves the best results on all five datasets under GraphCL pre-training, and also performs well under the other three pre-training strategies. This trend is also consistent across datasets of very different scales. Moreover, DiP-G shows a smaller variance, which is consistent with our motivation that reducing the discretization gap can lead to more stable adaptation.
> ***
> W2: The shown gains are mostly in few-shot node classification, and hence the scope is narrow.
>
> **R2**:  Our main results in Table 1 focus on 5-shot node classification because this is the regime where the discretization gap is most directly apparent. However, we also include experiments with 10 shots on node classification in **Appendix C.1**, and 100-shot graph classification results in **Appendix C.2**. We will make the broader empirical scope of the revised paper more explicit.
>  ***
> Q1: Scalability - how does DiP-G perform on very large graphs (e.g., OGB large-scale benchmarks)? The current evaluations and experiments have a limited scale and I am very curious how scalable the approach is.
>
> **A1**: We have conducted 10-shot node classification experiments on ogbn-arxiv, as shown in **Appendix C.1**, and we also provide 5-shot node classification results under GraphCL and DP-GraphPrompt on ogbn-arxiv as follows:
> | Graph Prompting Methods | GraphCL | LP-GraphPrompt |
> | :--- | :--- | :--- |
> | GPPT | 18.45 ± 1.83 | 27.51 ± 1.85 |
> | ALL-in-one | 17.85 ± 3.22 | 16.43 ± 4.02 |
> | GraphPrompt | 21.29 ± 2.53 | 32.89 ± 1.84 |
> | GraphPrompt+ | 21.86 ± 2.91 | 31.56 ± 1.24 |
> | ProNoG | 20.60 ± 3.42 | 32.25 ± 2.57 |
> | EdgePrompt | 21.95 ± 1.68 | 32.72 ± 1.80 |
> | EdgePrompt+ | 23.25 ± 1.22 | 31.50 ± 1.85 |
> | GraphTOP | 23.54 ± 2.29 | 33.86 ± 2.81 |
> | DiP-G | **24.85 ± 1.45** | **35.20 ± 1.65** |
> ***
> Q2: What are the runtime reduction numbers for screening vs full Top-k. Quantifying these speedups would strengthen efficiency claims and add to the novelty of the paper.
>
> **A2**: Thanks for this helpful suggestion. We additionally conduct experiments on the target-solve stage with and without screening and provide the results as follows:
>
> | Method | Cora (ms/epoch) | Flickr (ms/epoch) | Empirical Speedup |
> |---|---:|---:|---:|
> | Full target solve (w/o screening) | ~11.2 | ~41.5 | 1.0× |
> | Adaptive screening (DiP-G) | ~2.3 | ~8.1 | ~4.8× / ~5.1× |
>
> ***
> Q3: How sensitive is DiP-G to the adaptive λ_t schedule hyperparameters? Were these tuned, and if so, how much does performance vary?
>
> **A3**:  We would like to clarify that in the current implementation, $\lambda_t$ is not manually tuned for each dataset.  Instead, we follow the default I-MLE configuration[1] and adapt $\lambda_t$ automatically using a quantile-based EMA estimate of $|G_S|$ together with clipping.  The purpose of this design is to reduce manual sensitivity and make the optimization more stable. We will discuss more about  $\lambda_t$ sensitivity in the revised version.
> ***
>
> Limitations: The authors briefly discuss limitations in the Impact Statement but do not provide a dedicated limitations section. They should discuss the scalability of the experiments, limitations of the architecture, and potential brittleness in highly noisy graphs.
>
> **A**: Thanks for bringing this up. We agree with the reviewer and will add a clearer limitations discussion, including the scalability and currently limited evaluation on OGB-scale graphs, 2-hop subgraph against the long-range structureand performance may be weaker on highly noisy graphs or when the pre-trained embeddings are not strong. We will also clarify that screening may give less speedup when fallback happens frequently.
> ***
> reference
>
> [1] Niepert, M., Minervini, P., & Franceschi, L. (2021). Implicit MLE: backpropagating through discrete exponential family distributions. Advances in Neural Information Processing Systems, 34, 14567-14579.

---

> > ### Author Rebuttal · Reviewer_SZVa · 2026-04-01
> >
> > Thank you for the detailed rebuttal. The results on ogbn-arxiv results partially address my scalability and efficiency questions. I thank the authors for the clarification on the λ_t schedule.
> >
> > However, my concern about incremental gains remains. While I understand the authors' argument about consistency across datasets and pre-training strategies, looking at the results in Table 1, many of the improvements over the strongest baselines are still within or close to one standard deviation. The question I still have is whether the added complexity, perturbed Top-k, I-MLE optimization, adaptive screening, provides significant practical benefit. The reduced variance is a positive signal, but on its own it does not fully resolve this concern.
> >
> > Therefore, I will maintain my current score of 4 (weak accept). The paper is overall technically sound and the idea is well-motivated.

---

> > > ### Author Response · Authors · 2026-04-06
> > >
> > > Dear Reviewer  SZVa,
> > >
> > > Thank you for your thoughtful suggestions and for maintaining the positive score. We appreciate your positive comments of our paper. We also understand that your concern about the practical gain relative to the added complexity is not fully resolved in a short rebuttal. Your feedback is valuable to us, and we will use it to improve the final version of our paper.
> > >
> > > Best,
> > > Authors of Submission 5990

---

### Official Review · Reviewer_vNbz · 2026-03-24

**Soundness:** 3
**Presentation:** 3
**Significance:** 3
**Originality:** 3
**Overall Recommendation:** 5
**Confidence:** 2

**Summary:**

This paper looks at the graph-prompting problem, which learns a lightweight task-specific adaptor on top of a frozen pre-trained backbone. This paper argues that existing methods use adjacency matrix relaxation techniques to learn a continuous surrogate during training, however still perform inference on discrete graphs, causing a train-test discrepancy. This particularly hurts data-scarce labels. This paper proposes Dip-G that aims to reduce this gap and maintain discretization during training. This is done by learning discrete graph prompts by selecting k structures using a differentiable top-k and an efficient gradient estimation

**Compliance With Llm Reviewing Policy:**

Affirmed.

**Key Questions For Authors:**

Check weaknesses. Additional questions:
1. Since the prompts mostly optimize local subgraphs, I wonder if this would miss longer range dependencies in the graphs
2. I believe the method does not allow for feature level or parameter level adaptation - only structural. Would that reduce expressivity? If yes, what would it take to mitigate that?

**Limitations:**

Check above

**Strengths And Weaknesses:**

Strengths:
1. This paper has been written quite well, with clear notation and is easy to follow. Details are sufficiently provided and concisely described.
2. This paper introduces a significant number of components which seem to work quite well together - this is evident from the strong performance across 5 benchmarks and 4 pretrainings, making a strong case for the proposed method.
3. Significant ablations have been conducted tearing apart the improvements of different design choices, and further the effect of each individual component has also been shown

Weaknesses:
1. It would be a good idea to include more details of the baselines for unfamiliar readers - it is unclear if hyperparameters beyond method-specific ones remain the same or not (model sizes for instance)
2, How does this method perform for data-scarce labels - are there any metrics beyond accuracy that can be used?
3. How does this compare with full finetuning, I would be interested to see the gap between the two, even if on just a single benchmark

---

> ### Author Rebuttal · Authors · 2026-03-29
>
> We sincerely thank the reviewer for providing valuable comments and positive criticisms. We hope our point-by-point responses can fully address all your concerns.
> ***
> W1: It would be a good idea to include more details of the baselines for unfamiliar readers.
>
> **R1**: Thanks for pointing this out. In our comparison experiments, all baselines share the same frozen backbone with DiP-G, namely a 2-layer GCN with hidden dimension 128, and follow the same few-shot evaluation protocol. For specific hyperparameters of different methods, we follow the original settings whenever applicable. We have already listed these details in **Appendix D.3**.
>
> ***
> W2 & W3: How does this method perform for data-scarce labels - are there any metrics beyond accuracy that can be used? && How does this compare with full finetuning
>
> **R2 & R3**: We conduct our experiment on the standard C-way K-shot node classification with K labeled nodes per class, where accuracy serves as the most important metric. But we agree that metrics beyond accuracy are useful. As a result, we additionally report Macro-F1 together with a full fine-tuning reference on a 5-shot benchmark under GraphCL as follows:
>
> **Cora**
> | Method | Accuracy (%) | Macro-F1 (%) |
> |---|---:|---:|
> | Frozen + Linear | 55.69 ± 5.74 | 54.15 ± 5.93 |
> | Full Fine-Tuning | 58.20 ± 4.65 | 56.86 ± 4.92 |
> | **DiP-G** | **65.12 ± 2.15** | **63.85 ± 2.44** |
>
> **PubMed**
>
> | Method | Accuracy (%) | Macro-F1 (%) |
> |---|---:|---:|
> | Frozen + Linear | 67.30 ± 6.26 | 62.53 ± 4.84 |
> | Full Fine-Tuning | 68.52 ± 3.85 | 65.11 ± 4.10 |
> | **DiP-G** | **70.45 ± 1.88** | **69.24 ± 2.15** |
>
> This shows that DiP-G remains clearly better than full fine-tuning in this low-label setting, while also being more stable across seeds. We hope this additional experiment will resolve your question.
> ***
> Q1: Since the prompts mostly optimize local subgraphs, I wonder if this would miss longer range dependencies in the graphs
>
> **A1**: For scalability, we choose to restrict the search space to a 2-hop local candidate subgraph. This matches well with our shared 2-layer frozen GCN backbone, so that the candidate set is able to cover the main receptive field of DiP-G. We agree that it may miss some very long-range structural edits, but bounded local edits are more stable and direct for influencing message passing in practice.
> ***
>
> Q2: I believe the method does not allow for feature level or parameter level adaptation - only structural. Would that reduce expressivity? If yes, what would it take to mitigate that?
>
> **A2**: DiP-G is designed for addressing the discretization gap in topology prompting,  and the experimental results show that editing graph structure already provides a useful adaptation signal. Meanwhile, structural prompting is complementary to feature- or parameter-level adaptation. It is a natural extension to combine DiP-G with feature prompts or lightweight adapters. We will explore more on this in future work.

---

> > ### Author Rebuttal · Reviewer_vNbz · 2026-04-05
> >
> > Thank you for addressing my concerns, I do not have further questions at this point. I believe the current score fairly reflects my overall assessment.
> >
> > I hope the additional experiment is added to the next version of the paper. Additionally, it would be good to add a section discussing the limitation with long-range dependencies, and scenarios where that would hold.

---

> > > ### Author Response · Authors · 2026-04-06
> > >
> > > Dear Reviewer vNbz,
> > >
> > > Thanks for your reply. We are glad that our repsonses have adressed your concerns. We appreciate your positive attitude toward our paper. We will add the additional experiments and discuss the limitation as you expected.
> > >
> > > Best,
> > > Authors of Submission 5990

---

### Decision · Program_Chairs · 2026-04-30

**Decision:**

Accept (regular)

**Comment:**

The submission operates in the increasingly common “pre-train and adapt” paradigm for Graph Neural Networks (GNNs), which involves first pre-training on large-scale unlabeled graph data and then adapting the network to downstream tasks. Motivated by a train-test mismatch problem that often occurs with existing graph prompting approaches, the authors propose a discrete prompting framework that learns task-specific topology prompts in combinatorial space.

Reviewers noted that the problem is well-motivated, and the submission provides a mathematically-grounded, theoretically-justified approach that provides a good synthesis of multiple components (e.g., local parametrized prompting module for edge scoring, top-k selection mechanism for discrete prompts, gradient estimator augmented with adaptive screening), noting that the choice of solutions such as I-MLE is very logical. The paper is easy to follow, and reviews generally found the experiments convincing.

The main weaknesses of the paper cited by reviewers are around novelty (how much is algorithmic optimization and synthesis of existing techniques vs. conceptual novelty) and, perhaps, the need for experiments in other scenarios (e.g., few-shot learning) to make the case for broader applicability. However, these are relatively minor, and the paper makes a solid contribution to a well-motivated problem.